# Probabilistic Factorial Experimental Design for Combinatorial Interventions

**Divya Shyamal** [* 1 2]  **Jiaqi Zhang** [* 1 3]  **Caroline Uhler** [1 3]

## Abstract

A *combinatorial intervention*, consisting of multiple treatments applied to a single unit with potentially interactive effects, has substantial applications in fields such as biomedicine, engineering, and beyond. Given $p$ possible treatments, conducting all possible $2^p$ combinatorial interventions can be laborious and quickly becomes infeasible as $p$ increases. Here we introduce the *probabilistic factorial experimental design*, formalized from how scientists perform lab experiments. In this framework, the experimenter selects a dosage for each possible treatment and applies it to a group of units. Each unit independently receives a random combination of treatments, sampled from a product Bernoulli distribution determined by the dosages. Additionally, the experimenter can carry out such experiments over multiple rounds, adapting the design in an active manner. We address the optimal experimental design problem within an intervention model that imposes bounded-degree interactions between treatments. In the passive setting, we provide a closed-form solution for the near-optimal design. Our results prove that a dosage of $1/2$ for each treatment is optimal up to a factor of $1 + O(\ln(n)/n)$ for estimating any $k$-way interaction model, regardless of $k$, and imply that $O\left(kp^{3k}\ln(p)\right)$ observations are required to accurately estimate this model. For the multiround setting, we provide a near-optimal acquisition function that can be numerically optimized. We also explore several extensions of the design problem and finally validate our findings through simulations.

## 1. Introduction

In many domains, it is often of interest to consider the simultaneous application of multiple treatments/actions. For example, in cell biology, perturbing several genes is often necessary to induce a transition in cell state (Takahashi & Yamanaka, 2006). While a single treatment is constrained to a limited range of possible effects, a combinatorial intervention – comprising multiple treatments applied to the same unit – can result in a much wider array of outcomes. Much of this potential stems from the interactive effects between treatments, rather than merely the additive contributions of each. For example, perturbing paralogs (a pair of genes) can have a surprisingly larger effect than the sum of perturbing each gene individually, as one gene may compensate for the other, and only perturbing both simultaneously will effectively disrupt the pathway (Koonin, 2005). However, these interactions make the study of combinatorial interventions considerably more challenging than understanding single interventions alone, as each set of treatments may exhibit distinct interactions.

From the design perspective, the problem of testing combinatorial interventions to analyze the combined effects of various treatments is known as a factorial design (Fisher et al., 1966). Given $p$ possible treatments with large $p$, it is often infeasible to conduct all possible $2^p$ combinatorial interventions, which corresponds to a full factorial design. To address the scalability challenge, fractional factorial designs are introduced, which test only a subset of possible combinations. However, selecting this subset is difficult when prior knowledge is limited. Choosing a suboptimal subset may lead to a biased understanding of the experimental landscape. In addition, performing a large and specified subset of combinatorial interventions can be laborious and impractical, as each combination must be precisely assembled. In the perturbation example, this involves synthesizing a unique guide sequence for each combination that targets the specific genes involved (Rood et al., 2024). However, when the combination size is large, this becomes infeasible as a longer guide sequence may lack sufficient penetrance to effectively enter the targeted cells. To tackle these issues, we here formalize and study a scalable and unbiased approach to design factorial experiments.

Inspired by how scientists perform library designs in the

---

[*]Equal contribution  [1]Department of Electrical Engineering and Computer Science, MIT  [2]Department of Mathematics, MIT  [3]Eric and Wendy Schmidt Center, Broad Institute. Correspondence to: Jiaqi Zhang <viczhang@mit.edu>, Caroline Uhler <cuhler@mit.edu>.

*Proceedings of the $42^{nd}$ International Conference on Machine Learning*, Vancouver, Canada. PMLR 267, 2025. Copyright 2025 by the author(s).

lab (Yao et al., 2024), we introduce *probabilistic factorial experimental design*. In this framework, the experimenter selects a dosage for each possible treatment and applies it to a group of units. Each unit independently receives a random combination of treatments, sampled from a product Bernoulli distribution determined by the specified dosages. In the perturbation example mentioned above, this setup formalizes a high-multiplicity of infection (MOI) perturbation experiment (Yao et al., 2024), where multiple perturbations are applied at various MOI to a plate of cells, and each cell receives a combination of perturbations randomly. The introduction of a probabilistic design via dosages allow us to interpolate between an unbiased but expensive full factorial design and a relatively scalable but restricted fractional factorial design. By adjusting the dosages, we can effectively scale up a full factorial design by controlling the proportion of units receiving each combination in a realistic manner. Crucially, this approach remains unbiased as it does not require restricting the experiment to a predetermined subset of treatments. The question is then how to optimally design the dosages, e.g., in order to efficiently learn the interactions.

**Contributions.** Our contributions are summarized below.

- We propose and introduce the probabilistic factorial design, motivated by library design experiments in the lab (section 3.1). This setup assumes that treatments are randomly assigned to a group of units according to a prescribed dosage vector. It provides a scalable and flexible approach to implement factorial experiments, which we show to encapsulate both full and fractional factorial designs as special cases.

- Within this framework, we address the problem of optimal experimental design, which involves optimizing the dosage vectors based on a given objective. Our main focus is on learning the underlying combinatorial intervention model using a Boolean function representation assuming bounded-order interactions.

  - In the passive setting (section 4.2), we prove that assigning a dosage of $1/2$ to each treatment is near-optimal for estimating any $k$-way interaction model, leading to a sample complexity of $O(kp^{3k} \ln(p))$.
  - In the active setting (section 4.3), we introduce an acquisition function that can be numerically optimized and demonstrate that it is also near-optimal in theory.

- We explore several extensions to the design problem, including constraints on limited supply, heteroskedastic multi-round noise, and emulation of a target combinatorial distribution (section 5). Finally, we validate our theoretical findings through simulated experiments (section 6).

## 2. Related Works

**Factorial design.** Factorial experimental design has been extensively studied for its efficacy in evaluating multiple treatments simultaneously. Classical methods include full and fractional factorial designs (Fisher et al., 1966), and have been applied to various applications in biology, agriculture, and others (c.f., (Hanrahan & Lu, 2006)). Full factorial design considers all possible treatment combinations, where each treatment may have multiple levels (Deming & Morgan, 1993; Lundstedt et al., 1998; Dean & Voss, 1999). These experiments are sometimes conducted in multiple blocks, where each block is expected to have a controlled condition of external factors and contains one replicate of either all or partial combinations. When the number of total treatments increases, conducting such experiments quickly becomes infeasible. In these cases, fractional factorial design are preferred where a subset of carefully selected treatment combinations are tested (Gunst & Mason, 2009). A $2^{-m}$ fractional design is one where $2^{p-m}$ samples are used, each with a different combination (Box et al., 1978). These combinations are carefully selected to minimize aliasing. Aliasing occurs when, for the combinations selected, the interactions are linearly dependent (Gunst & Mason, 2009; Mukerjee & Wu, 2007). In a full factorial design, there is linear independence, so there is no confounding when the model is fit. In a fractional design, some aliasing will always occur in a full-degree model; however, methods proposed in the literature select combinations such that the aliasing of important effects (i.e. degree-1 terms) does not occur (Gunst & Mason, 2009). With little prior knowledge, it is common to assume that low-order effects are more important than higher-order interactions and select designs to focus on low-order effects (Cheng, 2016). With a low-degree assumption, aliasing can be avoided entirely. Fractional designs can be classified by their *resolution* (denoted by $R$), which determines which interactions can be potentially confounded. For example, a Resolution V fractional design eliminates any confounding between lower than degree-3 interactions, appropriate for degree-2 functions (Montgomery, 2017). Of particular interest in literature are *minimum aberration designs*, which minimize the number of degree-$l$ terms aliased with degree-$R - l$ terms (Fries & Hunter, 1980; Cheng, 2016). However, scalability to high-dimensional problems remains a challenge, and efficient sampling methods such as Bayesian optimization are proposed (Mitchell et al., 1995; Kerr, 2001; Chang & Cheng, 2018).

The probabilistic setting proposed in this paper serves as a flexible realization of a factorial design that automatically generates a design resembling either a full factorial or a fractional factorial design, depending on the selected dosages. We formally discuss this in section 3.1.

**Learning combinatorial interventions.** Modeling combinatorial interventions is crucial for understanding their interactions and designing experiments. There are multiple ways to model such interventions, often by imposing structures that relate different combinations. For example, the Bliss independence (Bliss, 1939) and Loewe additivity (Loewe, 1926) models are commonly used to describe additive systems where no interactions between treatments are assumed.

An alternative approach is to use a structural causal model (SCM) and the principal of independent causal mechanisms (Eberhardt, 2007; Eberhardt & Scheines, 2007). In particular, this assumes that (1) each single-variable intervention alters the dependency of that variable on its parent variables according to the SCM, and (2) a combinatorial intervention modifies each involved variable according to its respective single-variable intervention and then combines these changes in a factorized joint distribution. Within this framework, various types of interventions, including do-, hard-, and soft-interventions, can be defined (e.g., (Correa & Bareinboim, 2020; Zhang et al., 2023)). However, similar to the Bliss independence and Loewe additivity models, SCM-based approaches cannot capture interactions between treatments.

To model such interactions, one can use a generalized surface model, which can be instantiated via polynomial functions (Lee, 2010) or Gaussian processes (Shapovalova et al., 2022). Alternatively, Boolean functions provide another modeling framework (Agarwal et al., 2023a), where theoretical tools such as the Fourier transform can be leveraged (O'Donnell, 2008). Agarwal et al. has employed this approach, where sparsity and rank constraints are used to enforce structural assumptions on combinatorial interactions. In this paper, we also utilize Boolean functions, where we demonstrate their close relationship with generalized surface models. We show that interactions can be read-off from Fourier coefficients, allowing us to formalize assumptions about the degree of interactions.

## 3. Setup and Model

In this section, we propose and define the setup of probabilistic factorial experimental design. We then introduce the outcome model we use to model combinatorial interventions and discuss its applicability to model interactions.

### 3.1. Probabilistic Factorial Design Setup

Consider $p$ possible treatments with $2^p$ total combinatorial interventions. In a *probabilistic factorial experimental design*, the experimenter chooses a vector of *dosages*, denoted by $\mathbf{d} = (d_1, \ldots, d_p) \in [0,1]^p$, and applies the treatments at this level to $n$ homogenous units. For sim-

plicity, we consider no interference between units, where each unit independently receives a combinatorial intervention at random. Denote the intervention associated with unit $m \in [n] = \{1, \ldots, n\}$ by $\mathbf{x}_m \in \{-1, 1\}^p$, where $x_{m,i} = 1$ if and only if it receives a combinatorial intervention that contains treatment $i$. Here $\mathbf{x}_m$ is randomly sampled according to a product Bernoulli distribution according to $\mathbf{d}$, where

$$x_{m,i} = \begin{cases} 1 & \text{with probability } d_i, \\ -1 & \text{with probability } 1 - d_i. \end{cases} \quad (1)$$

The experimenter can carry out such experiments for $T$ times, with different dosages $\mathbf{d}_1, \cdots, \mathbf{d}_T$, potentially in a sequential and adaptive manner. In combinatorial perturbation example in section 1, the dosage vector formalizes the multiplicity of infection of each considered perturbation.

Note that this setup reduces to traditional two-level factorial design (Fisher et al., 1966) by choosing $\mathbf{d} \in \{0,1\}^p$. In particular, for any combinatorial intervention consisting of treatments in $S \subseteq [p]$, setting $d_i = 1$ if $i \in S$ or else $d_i = 0$ gives rise to all units receiving $S$. Allowing for continuous $\mathbf{d} \in [0,1]^p$ generalizes this setup by enabling the allocation of units to different combinatorial interventions in a realistic and effective manner controlled by $\mathbf{d}$.

### 3.2. Outcome Models for Combinatorial Interventions

Under this setup, we are interested in estimating the average treatment effect of combinatorial interventions. For unit $m$, we observe its treatment assignment $\mathbf{x}_m$ and outcome $y_m \in \mathbb{R}$. We adopt the outcome model proposed by Agarwal et al. (2023b), where $y_m$ corresponds to a noisy observation of a *real-valued Boolean function* $f : \{-1, 1\}^p \to \mathbb{R}$, i.e.,

$$y_m = f(\mathbf{x}_m) + \epsilon_m.$$

Here we assume $\epsilon_m$ is independent among different units and is normally distributed with mean zero and variance $\sigma^2$. This model choice has the flexibility of allowing for interactions between arbitrary sets of treatments, as we illustrate below.

The class of real-valued Boolean functions admits a representation via the *Fourier basis*

$$\{\phi_S(\mathbf{x}) = \prod_{i \in S} x_i \mid S \subseteq [p]\}$$

by

$$f(\mathbf{x}) = \sum_{S \subseteq [p]} \beta_S \phi_S(\mathbf{x}).$$

Here $\beta_S = \frac{1}{2^p} \sum_{\mathbf{y} \in \{-1,1\}^p} f(\mathbf{y})\phi_S(\mathbf{y})$ (see Appendix A for details). The Fourier coefficients are interpretable in the sense that the polynomial instantiation of the generalized

response surface model (Lee, 2010) can be expressed in this form, where all the $k$-way interactions are captured by

$$\{\beta_S \mid S \subseteq [p], |S| \leq k\}.$$

In particular, the generalized response surface model can be written as follows.

**Polynomial Instantiation.** To capture the nonlinear interactions between treatments, we can model the outcome of combinatorial intervention $\mathbf{x}$ via

$$f(\mathbf{x}) = \sum_{i=1}^{p} \alpha_i \mathbb{1}_{x_i=1} + \sum_{i,j=1}^{p} \alpha_{ij} \mathbb{1}_{x_i=x_j=1} + \dots,$$

where $\alpha_S$ represents the contribution in the final outcome by the interaction among treatments in $S$. This model can be represented via the Fourier representation (see Appendix A for details), where

$$\beta_S = \sum_{S \subseteq T} \frac{\alpha_T}{2^{|T|}}. \tag{2}$$

In a bounded-order interaction model, $\alpha_S = 0$ for large $|S|$. In particular, if $\alpha_S = 0$ for $|S| > k$, then $\beta_S = 0$ for $|S| > k$ according to Eq. (2). This motivates us to make the following assumptions on the Fourier coefficients.

**Assumption 3.1** (Bounded-order interactions). *The outcome model exhibits bounded-order interactions, i.e., there exists $k = o(p)$ such that*

$$\beta_S = 0 \quad \text{if} \quad |S| > k.$$

We also assume that $\beta$ is bounded in $L_2$ norm.

**Assumption 3.2.** (Boundedness of $\beta$) *There exists a constant $B$ such that $\|\beta\|_2 \leq B$.*

## 4. Optimal Experimental Design

In this section, we focus on optimal experimental design for learning the outcome model $f$. We consider extensions of these results in section 5. For the objective of learning $f$, we provide near-optimal design strategies for the choice of dosages $\mathbf{d}$ in both passive and adaptive scenarios. We start by introducing the estimators of $f$. All formal proofs in this section are deferred to Appendix B.

### 4.1. Estimators

Estimating the Fourier coefficients $\beta$ accurately in turn gives an accurate estimate of $f$, as $\|\hat{f}(\mathbf{x}) - f(\mathbf{x})\|_2 \leq \|\hat{\beta} - \beta\|_2$, where $\hat{f}(\mathbf{x}) = \sum_{S \subseteq [p]} \hat{\beta}_S \phi_S(\mathbf{x})$. Therefore, it suffices to focus on $\hat{\beta}$.

Denote the collected dataset as $\mathbb{D} = \{(\mathbf{x}_m, y_m) \mid m \in [n]\}$. Let the design matrix be $\mathcal{X} \in \mathbb{R}^{n \times K}$ with $K = \sum_{i=0}^{k} \binom{p}{k}$.

The columns of $\mathcal{X}$ corresponds all possible combinations (including size $\leq 1$) with interactions, i.e., $S \subseteq [p]$ with $|S| \leq k$. The $m$-th row of $\mathcal{X}$ corresponds to the Fourier characteristics of the observed combination $\mathbf{x}_m$ with

$$\mathcal{X}_{m,S} = \phi_S(\mathbf{x}_m) = \prod_{i \in S} x_{m,i}.$$

Given that $\mathcal{X}$ is randomly drawn according to the dosages $\mathbf{d}$ and its columns are correlated, it is possible that it is ill-conditioned for a standard linear regression estimator. Therefore, in order to control the estimation error, we use a truncated ordinary least squares (OLS) to estimate $\beta$:

$$\hat{\beta} = \begin{cases} (\mathcal{X}^\top \mathcal{X})^{-1} \mathcal{X}^\top Y & \text{if } \sum_{i=1}^{K} \lambda_i (\mathcal{X}^\top \mathcal{X})^{-1} \leq \frac{B^2}{\sigma^2}, \\ \mathbf{0} & \text{otherwise.} \end{cases}$$

Here $\lambda$ denotes the eigenvalues of $\mathcal{X}^\top \mathcal{X}$ and $Y$ is the vector by stacking $y_m$ with $m \in [n]$. Note that this results in a null estimator when the eigenvalues are small. We use this to demonstrate the key ideas of our analysis in a simpler form In practice, when $\mathcal{X}$ is ill-conditioned, alternative estimators such as ridge regression can be used, where similar theoretical results can be derived (see Appendix B for details). The truncated OLS estimator satisfies the following property, which we utilize in our analysis.

**Lemma 4.1.** *Given a fixed design matrix $\mathcal{X}$, the truncated OLS estimator satisfies*

$$\min\{\sum_{i=1}^{K} \frac{\sigma^2}{\lambda_i(\mathcal{X}^\top \mathcal{X})}, \|\beta\|_2^2\} \leq$$
$$\mathbb{E}_Y\left[\|\hat{\beta} - \beta\|_2^2\right] \leq \min\{\sum_{i=1}^{K} \frac{\sigma^2}{\lambda_i(\mathcal{X}^\top \mathcal{X})}, B^2\}. \tag{3}$$

### 4.2. Passive Setting

In this scenario, the experimenter decides the choice of the dosages $\mathbf{d}$ in a prospective fashion without considering any data collected in the past. This is in contrast to an active design, where collected data are utilized to decide the current design. Note that the first round of any active setting reduces to the passive scenario, as there is no collected data.

Suppose we have a budget of $n$ units. To select $\mathbf{d}$ such that we can obtain the most accurate estimate of $\beta$ after observing these units, it is natural to optimize the following objective:

$$\mathbb{E}_{\mathbb{D}}\left[\|\hat{\beta} - \beta\|_2^2\right]. \tag{4}$$

We show that this objective has a closed-form near-optimal solution of $\mathbf{d} = (1/2, \cdots, 1/2)$, regardless of the order of the interactions.

**Theorem 4.2.** *For the truncated OLS estimator, $\mathbf{d} = (1/2, \cdots, 1/2)$ is optimal up to a factor of $1 + O\left(\frac{\ln(n)}{n}\right)$*

*with respect to Eq. (4). In addition, the minimizer of Eq. (4) lies in an $l_\infty-$norm ball centered on the half dosage with radius $O\left(\sqrt{\frac{\ln(n)}{n}}\right)$.*

Note that with the half dosage, the probability of observing any particular combinatorial intervention $S \subseteq [p]$ is $2^{-p}$. Therefore in the passive setting, it is always optimal to evenly administer every treatment so that the observed combinatorial interventions follow a uniform distribution.

*Proof sketch.* In Lemma 4.1, we show how to bound the expectation of the error $\|\hat{\boldsymbol{\beta}} - \boldsymbol{\beta}\|_2^2$ with respect to randomness in the outcome $Y$. To obtain the optimal dosage for Eq. (4), we need to additionally take expectation with respect to the randomness of $\mathcal{X}$, which is where the dosages enter as the combinatorial interventions $\mathbf{x}$ are sampled from Eq. (1). Note that $\mathbb{E}[\mathcal{X}^\top \mathcal{X}] = n \cdot \Sigma(\mathbf{d}) \in \mathbb{R}^{K \times K}$, where

$$\Sigma(\mathbf{d})_{S,S'} = \prod_{i \in S \Delta S'} (2d_i - 1), \qquad (5)$$

for any $S, S' \subseteq [p]$ such that $|S|, |S'| \le k$.[1]

*Intuition of the optimality of half dosages.* For the standard OLS estimator, the expected squared error is

$$\sigma^2 \sum_{i=1}^{K} \frac{1}{\lambda_i(\mathcal{X}^\top \mathcal{X})}.$$

If we directly swap $\mathcal{X}^\top \mathcal{X}$ with its expected value, then we need to minimize

$$\sum_{i=1}^{K} \frac{1}{\lambda_i(\Sigma(\mathbf{d}))}.$$

Note that $\mathrm{tr}(\Sigma(\mathbf{d})) = K$ for all $\mathbf{d}$. Therefore, by the Cauchy-Schwarz inequality, $\sum_{i=1}^{K} \lambda_i(\Sigma(\mathbf{d}))^{-1}$ is minimized if and only if $\lambda_i(\Sigma(\mathbf{d})) = 1$ for all $i \in [K]$, which is satisfied when $\Sigma(\mathbf{d}) = \mathbf{I}_K$ and $\mathbf{d} = (1/2, \ldots, 1/2)$.

To formally show that the expected error is optimized with half dosages, we can use a concentration result for $\mathcal{X}^\top \mathcal{X}$ which can be obtained using an $\epsilon$-net argument (Vershynin, 2018) and Hoeffding's inequality. However, the eigenvalues of $\mathcal{X}^\top \mathcal{X}$ enters the error computation through the denominators, which makes the computation difficult. In particular, $\mathbb{E}(\sum_{i=1}^{K} \lambda_i(\mathcal{X}^\top \mathcal{X})^{-1})$ cannot be bounded due to exploding terms when $\lambda_{\min}(\mathcal{X}^\top \mathcal{X})$ approaches zero. We resolve this difficulty by utilizing the bounds in Lemma 4.1. For

---

[1]Here $\Delta$ denotes the disjunctive union, i.e., $S \Delta S' = (S \cup S') \setminus (S \cap S')$.

$\mathbf{d} = (1/2, \ldots, 1/2)$, we use the upper bound to show that

$$\mathbb{E}_\mathbb{D}\left[\|\hat{\boldsymbol{\beta}} - \boldsymbol{\beta}\|_2^2\right] \le \frac{K\sigma^2}{n(1-\delta)} + B^2 \exp\left(K \ln 9 - \frac{n\delta^2}{8K^2}\right)$$

for any $0 < \delta < 1$. For $\mathbf{d}$ such that $\max_i |2d_i - 1| > 0$, we use the lower bound to show that,

$$\mathbb{E}_\mathbb{D}\left[\|\hat{\boldsymbol{\beta}} - \boldsymbol{\beta}\|_2^2\right] \ge \left(1 - 2\exp\left(K \ln 9 - \frac{\delta^2}{8K^2}\right)\right) \cdot$$
$$\min\{\frac{\sigma^2}{n(1 - \max_i |2d_i - 1| + \delta)} + \frac{\sigma^2(K-1)}{n(1+\delta)}, \|\beta\|_2^2\},$$

for any $\delta > 0$. By choosing $\delta = (2 \ln n/n)^{1/2}$, we obtain that $\mathbf{d} = (1/2, \ldots, 1/2)$ is optimal up to a factor of $1 + O(\frac{\ln(n)}{n})$. $\square$

As a corollary of the proof for Theorem 4.2, we can show the error of estimating $\boldsymbol{\beta}$ decays with a rate of $n^{-1}$.

**Corollary 4.3.** *With $\mathbf{d} = (1/2, \ldots, 1/2)$, there is*

$$\mathbb{E}_\mathbb{D}\left[\|\hat{\boldsymbol{\beta}} - \boldsymbol{\beta}\|_2^2\right] \le \frac{2K\sigma^2 + 1}{n}$$

*for $n > n_0$, where $n_0 = O\left(K^3 \ln K\right)$.*

Therefore in order to estimate a $k$-way interaction model correctly, $O(K^3 \ln(K)) = O\left(kp^{3k} \ln(p)\right)$ samples suffice.

### 4.3. Active Setting

In this setting, the experimenter decides the choice of the dosages $\mathbf{d}$ sequentially in multiple rounds, where the observations from previous rounds can be used to inform the choice of dosage. Note that as discussed in Section 4.2, the first round of the active setting degenerates to the passive setting, where the optimal choice is $\mathbf{d} = (1/2, \ldots, 1/2)$.

Consider round $T > 1$. Denote $\mathbb{D}_t$ as the collected data and let $\mathcal{X}_t$ be the design matrix obtained by $\mathbb{D}_t$ at round $t \le T$. The goal is to minimize the following objective

$$\mathbb{E}_{\mathbb{D}_T}\left[\|\hat{\boldsymbol{\beta}} - \boldsymbol{\beta}\|_2^2 \mid \mathbb{D}_1 \cup \ldots \mathbb{D}_T\right]. \qquad (6)$$

In this scenario, we can not obtain a closed form solution as the optimal choice of $\mathbf{d}$ depends on pre-collected $\mathbb{D}_1 \cup \ldots \mathbb{D}_{T-1}$, which can be arbitrary. However, we show it is possible to derive a near-optimal objective that can be easily computed and numerically optimized.

**Theorem 4.4.** *The following choice of dosage:*

$$\mathbf{d}_T = \underset{\mathbf{d} \in [0,1]^p}{\arg\min} \sum_{i=1}^{K} \frac{1}{\lambda_i\left(\Sigma(\mathbf{d}) + \frac{1}{n}\sum_{t=1}^{T-1} \mathcal{X}_t^\top \mathcal{X}_t\right)} \qquad (7)$$

**Algorithm 1** Active probabilistic factorial experimental design.

---
1: Initialize $\mathbb{X}^\top \mathbb{X} = \mathbf{0}_{M \times M}$.
2: **for** $t = 1$ to $T$ **do**
3:     **if** $t = 1$ **then**
4:        set $\mathbf{d} = (1/2, \ldots, 1/2)$;
5:     **else**
6:        set $\mathbf{d}_t = \operatorname*{argmin}_{\mathbf{d} \in [0,1]^p} \sum_{i=1}^K \frac{1}{\lambda_i \left( \Sigma(\mathbf{d}) + \frac{1}{n} \sum_{i=1}^{t-1} \mathcal{X}_i^\top \mathcal{X}_i \right)}$.
7:     **end if**
8:    Gather $n$ observations according to Eq. (1) and form design matrix $\mathcal{X}_t$.
9:    Update $\mathbb{X}^\top \mathbb{X} \leftarrow \mathbb{X}^\top \mathbb{X} + \frac{1}{n} \mathcal{X}_t^\top \mathcal{X}_t$
10: **end for**
11: Return estimated $\boldsymbol{\beta}$ using all observations.

---

*is optimal up to a factor of $1 + O\left(\frac{\ln(n)}{n}\right)$ with respect to Eq.* (6).

In practice, we solve for $\mathbf{d}_T$ by numerically optimizing the objective in Eq. (7) using the SLSQP solver in Scipy (Virtanen et al., 2020). The number of iterations for the optimizer to converge is roughly $O(p^3)$, and the complexity of each iteration is $O(nK^2 + K^3)$ (where the first term comes from the matrix multiplication of $\mathcal{X}^T\mathcal{X}$ and the second term comes from computing the eigenvalues of $\Sigma(\mathbf{d})$). Recall the definition of $K$ to be the number of interactions under consideration, i.e. $K = \sum_{i=0}^k \binom{p}{i} = O(p^k)$ for small $k$. Therefore, the overall complexity is $O(np^{3k+3} + p^{6k+3})$ for small $k$. In practice, we may recommend using a proxy, which only involves the inverse of the minimum eigenvalue: $\mathbf{d_T} = \operatorname{argmin}_{\mathbf{d} \in [0,1]^p} \frac{1}{\lambda_{\min}\left(\Sigma(\mathbf{d}) + \frac{1}{n}\sum_{t=1}^{T-1} \mathcal{X}_t^\top \mathcal{X}_t\right)}$. We found that numerically optimizing this was significantly faster and that the solver was consistently accurate. While the complexity computed above should be the same for this approach, in practice it takes many less iterations to converge. We summarize the procedure for the active setting in Algorithm 1.

# 5. Extensions

In this section, we consider several extensions and discuss how the design policy changes in different scenarios.

## 5.1. Limited Supply Constraint

Here, we consider the case where we have additional constraint on the possible dosages $\mathbf{d}$:

$$\sum_{i=1}^p d_i \leq L, \quad \text{for some } 0 < L < \frac{p}{2}. \tag{8}$$

We assume $L < \frac{p}{2}$, as otherwise $\mathbf{d} = (1/2, \ldots, 1/2)$ is feasible and therefore optimal. This case is inspired by a

setting where we have supply constraints on treatments, or where we do not want to assign a unit too many treatments at once. Note that the constraint implies that the expected number of treatments assigned to a unit is at most $L$.

In the passive setting, we derive a closed-form near-optimal dosage for the pure-additive model, i.e. $k = 1$ in Assumption 3.1. This result requires understanding of the spectrum of $\Sigma(\mathbf{d})$. In the no-interaction case, we are able to derive the characteristic polynomial for $\Sigma(\mathbf{d})$, which becomes difficult when $k > 1$. However, empirical results show that the result, which we now state, to hold for $k > 1$ as well (see section 6).

**Theorem 5.1.** *For the additive model with $k = 1$, among the dosages that satisfy the constraint in Eq.* (8), *the uniform dosage $\mathbf{d}$ with $d_i = \frac{L}{p}$ for all $i \in [p]$ is optimal up to a factor of $1 + O\left(\frac{\ln(n)}{n}\right)$ with respect to Eq.* (9).

For non-additive models and the active setting, we note that Theorem 4.4, where the feasible region of $\mathbf{d}$ is modified according to Eq. (8), to still hold. Therefore, although no closed-form solution can be derived, we can still obtain a near-optimal solution via numerical optimization.

## 5.2. Heteroskedastic Multi-round Case

Our results can easily extend to the scenario where the noise in the outcomes varies by round. This case might be relevant when different rounds of experiments have systematic batch effects, e.g., if they are collected within different labs.

Assume that in round $t$, the variance of the observed outcome noise is $\sigma_t^2$. Note that in this setting, $\mathbf{d} = (1/2, \ldots, 1/2)$ is still near-optimal for the first round. However, the optimal choice of dosage at round $T$ becomes

$$\mathbf{d}_T = \operatorname*{argmin}_{\mathbf{d} \in [0,1]^p} \sum_{i=1}^K \frac{1}{\lambda_i \left( \frac{1}{\sigma_T^2} \Sigma(\mathbf{d}) + \frac{1}{n} \sum_{t=1}^{T-1} \frac{1}{\sigma_t^2} \mathcal{X}_t^\top \mathcal{X}_t \right)}$$

where we now scale the observations at round $t$ by $\frac{1}{\sigma_t}$ and use the truncated OLS estimator on this modified dataset (Eq. (6)), following a weighted least squares approach.

## 5.3. Limited Intervention Cardinality

Consider the scenario where the set of possible treatments that can be applied has limited cardinality:

$$\|\mathbf{d}\|_0 \leq L, \quad \text{for some } 0 < L < p.$$

Suppose that $d_i \neq 0$ for $i \in D$, where the cardinality $|D|$ is bounded by $L$. Then it holds that $\mathcal{X}_{:,S} = (-1)^{|S \setminus D|} \mathcal{X}_{:,S \cap D}$. Therefore the design matrix can be written as

$$\mathcal{X} = \mathcal{X}_D \Gamma_D$$

where $\mathcal{X}_D$ denotes the submatrix of $\mathcal{X}$ corresponding to columns $\mathcal{X}_{:,S}$ with $S \subseteq D$ and $\Gamma_D$ consists of one-hot vectors as columns. In this case, we may estimate $\boldsymbol{\beta}$ only up to $\Gamma_D \boldsymbol{\beta}$, e.g., using the following truncated OLS estimator

$$
\begin{cases}
(\mathcal{X}_D^\top \mathcal{X}_D)^{-1} \mathcal{X}_D^\top Y & \text{if } \sum_{i=1}^{K} \lambda_i (\mathcal{X}_D^\top \mathcal{X}_D)^{-1} \leq \frac{B^2}{\sigma^2}, \\
\mathbf{0} & \text{otherwise.}
\end{cases}
$$

Note that this has a form similar to $\hat{\boldsymbol{\beta}}$, where using similar arguments as in Section 4, we can show that $d_i = 1/2$ for $i \in D$ is near optimal. Thus, in the passive setting, the near-optimal strategy becomes selecting a subset of treatments $D$ with $|D| \leq L$ and setting $d_i = 1/2$ for $i \in D$ and $d_i = 0$ for $i \notin D$. As the estimator for $\Gamma_D \boldsymbol{\beta}$ directly estimates entries $\boldsymbol{\beta}_S$ of $\boldsymbol{\beta}$ with $S \subseteq D$, one can select $D$ based on prior preference of which coefficients of $\boldsymbol{\beta}$ are of interest.

### 5.4. Emulating a Target Combinatorial Distribution

We consider a different problem that explores the possibility of emulating a target distribution of combinatorial interventions with one round of probabilistic factorial design.

Formally, let $q$ be an arbitrary distribution over all possible combinatorial interventions, we are interested in approximating $q$ with choices of $\mathbf{d}$. Denote $p_{\mathbf{d}}$ as the distribution over combinatorial interventions induced by dosage $\mathbf{d}$. We use KL divergence $D(q \,||\, p_{\mathbf{d}})$ to measure the approximation error. To optimize over $\mathbf{d}$, note that $p_{\mathbf{d}}$ is a product distribution and we have

$$
D(q \,||\, p_{\mathbf{d}}) = H(q) - \sum_{i=1}^{p} q_i \log(d_i) - (1 - q_i) \log(1 - d_i),
$$

where $q_i = \sum_{\mathbf{x}_i = 1} q(\mathbf{x}_i)$ is the marginal distribution of receiving treatment $i$ under the target distribution, and $H(\cdot)$ denotes the entropy. Minimizing this equality quickly obtains $d_i = q_i$, which indicates choosing $\mathbf{d}$ based on the marginals of the target distribution. The minimal approximation error is then

$$
H(q) - H(q_1 \otimes \dots q_p),
$$

which means we can emulate a target distribution well if it is closed to a product distribution.

## 6. Experiments

We conduct experiments to validate our theoretical results, as well as show a comparison to fractional factorial design, using simulated data. We generate the outcome model $f$ by sampling the Fourier coefficients from the uniform distribution, i.e., $\boldsymbol{\beta} \sim \mathcal{U}(-1, 1)^K$. We noise the outcomes with standard Gaussian noise. In each of the following simulations, we keep $\boldsymbol{\beta}$ constant through all iterations of each run. Further details and the code repository can be found in Appendix D.

### 6.1. Comparison to Fractional Factorial Design

Here we compare the half dosage versus a partial factorial design in the passive setting. We generate a degree-1 Boolean function with $p = 8$. We use a $2^{8-2}$ Resolution $V$ design with 64 samples for each approach.

The fractional design returns a mean squared error of $0.14 \pm 0.062$, where the half dosage gives $0.16 \pm 0.078$ (averaged over 300 trials and with $\pm 1$ std). With fewer samples, the careful selection of combinations will make a difference, so the fractional design can outperform the half dosage. But in many cases, especially in biological applications, careful selection of combinations is not possible which is why the much more flexible dosage design is preferable, as it enables the administration of an exponential number of combinations by choosing a linear number of dosages.

However, in the active setting, the optimal dosage can outperform a fractional design. This is discussed further in Section 6.3.

### 6.2. Passive Setting

In Theorem 4.2, we show that $\mathbf{d} = (1/2, \dots, 1/2)$ is optimal up to a factor of $1 + O(\frac{\ln(n)}{n})$. Empirically, our validations build on the comparison of estimation error between half dosages and randomly sampled dosage vectors. We consider two different ways to generate dosages in this comparison, as described below.

**Simulation 1.** Here, we investigate the approximation of $\hat{\beta}$ achieved by different dosages $\mathbf{d}$ based on their $l_\infty$-distances from the $\frac{1}{2} := (1/2, \dots, 1/2)$, i.e., $\left\| \mathbf{d} - \frac{1}{2} \right\|_\infty$. We consider distances ranging from 0 to .4, where we sample 100 different dosage vectors at each distance. For each dosage, we generate 20 sets of observations and regress on each.

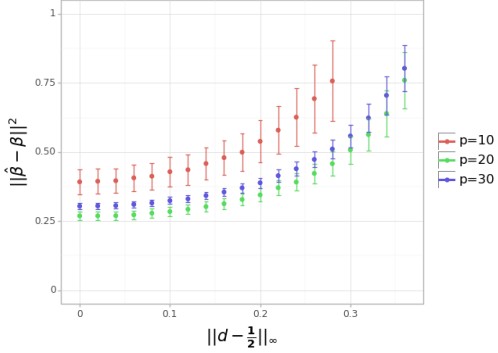

*Figure 1.* **Simulation 1.** Average $\|\hat{\boldsymbol{\beta}} - \boldsymbol{\beta}\|_2^2$ over 2000 different observation sets generated from 100 different dosages at each given distance. The bars correspond to $\pm.5$ std over the 2000 observations. The curves are generated with values $p = 10, k = 2, n = 200$; $p = 20, k = 2, n = 1000$; and $p = 30, k = 2, n = 1000$.

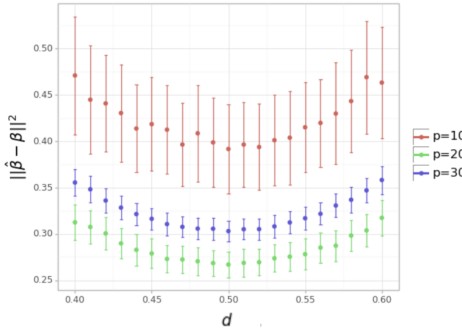

*Figure 2.* **Simulation 2.** Approximation error of uniform dosages. Bars correspond to $\pm.5$ std over $500$ observations per dosage. The curves are generated with values $p = 10, k = 2, n = 200$; $p = 20, k = 2, n = 1000$; and $p = 30, k = 2, n = 1000$.

We show these results for three different sets of $p, k,$ and $n$ in Figure 6.2. These values are chosen such that the ratio $K/n$ is kept approximately constant under different number of total treatments, following Corollary 4.3: $p = 10, k = 2, n = 200$; $p = 20, k = 2, n = 1000$; and $p = 30, k = 2, n = 1000$.

**Simulation 2.** Here, we only consider dosages where each treatment is administered at the same dosage, which we refer to as a *uniform dosage*. We consider dosage values ranging from $.4$ to $.6$, and generate $500$ different observation sets for each dosage. We show the approximation error of $\hat{\boldsymbol{\beta}}$ against the dosage value in Figure 2 for the three different sets of $p, k,$ and $n$ used in simulation 1.

**Results.** In simulation 1, we see that the approximation error is generally increasing in $\left\| \mathbf{d} - \frac{1}{2} \right\|_\infty$. Even with relatively small $n$ (on the scale of $O(K)$, rather than $\text{poly}(K)$ in Corollary 4.3), we see that the half dosage seems to be optimal. In simulation 2, we again see that the half dosage exhibits optimality, with $U-$shaped curves dipping at $.5$.

### 6.3. Active Setting

Here, we carry out 10 sequential experimental rounds. We compare our proposed choice of dosage in Theorem 4.4, which we refer to as optimal, to two baselines. The first baseline, referred to as random, randomly chooses a dosage from $\mathcal{U}(0,1)^p$ at each round. The second baseline, referred to as half, chooses the dosage of $\frac{1}{2}$ at each round. On synthetic data, we find that optimal and half perform similarly when $n$ is relatively large, while clearly outperforming random (Figure 3). In settings where $n$ is relatively small, optimal outperforms half (Figure 4). We also add a partial design baseline, referred to as partial (a Resolution $V$ $2^{5-1}$ design), in the small $p$ setting. In earlier rounds, we see optimal performs the best, and partial catches up after sufficiently many rounds.

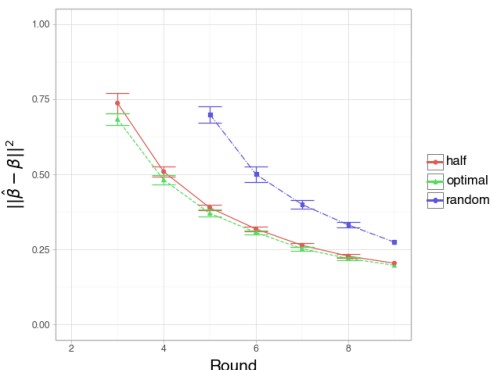

*Figure 3.* **Active setting with relatively large** $n$. Results are averaged over $20$ trials, where $p = 15, k = 2, n = 75$. We limit the $y-$axis to 1, focusing on later rounds when the approximation error is small. Bars correspond to $\pm.2$ std.

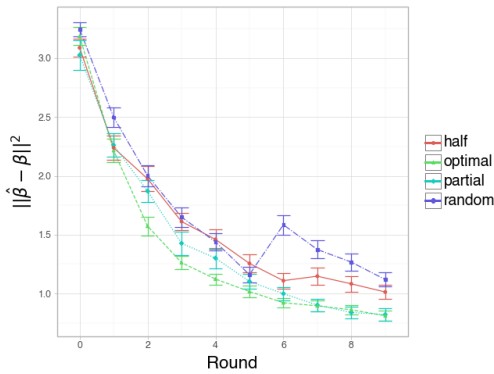

*Figure 4.* **Active setting with relative small** $n$**, high noise.** Results are averaged over $50$ trials, where $p = 5, k = 1, n = 16, \sigma = 5$. Bars correspond to $\pm.1$ std.

**Results.** We see that random performs consistently worse that optimal and half. For high $n$ (compared to $K$), the difference between optimal and half is marginal (as seen in Figure 3). However, when $n$ is small, there is a noticeable gap between optimal and half. In the case where there are not many samples (compared to features) per round, we find that the optimal acquisition strategy more clearly outperforms the half strategy. This is because when we have a smaller number of samples, we will need to "correct" as the distribution of combinations will be more lopsided and further away from the uniform distribution. Similarly, this is why optimal can outperform partial in a multiple-round setting, though it may be subpar in a single round. Therefore, in scenarios where each round has few samples, we think it is worth computing the optimal acquisition dosage. When we have a large $n$ relative to $p$, the half strategy and optimal strategy perform very similarly.

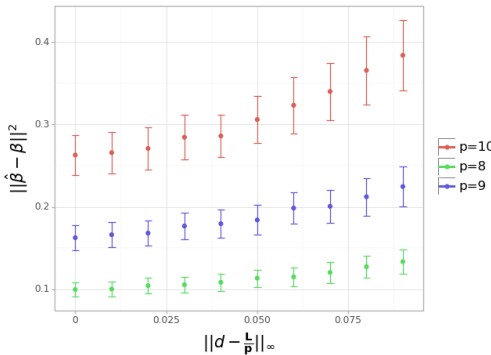

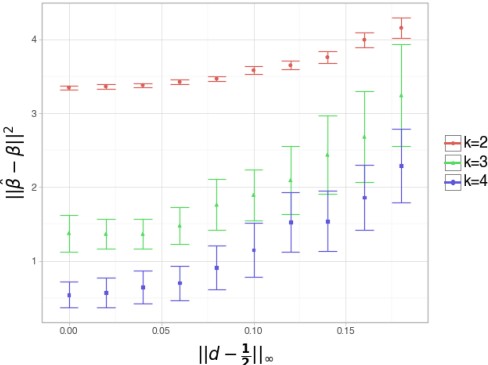

*Figure 5.* **Limited Supply Constraint.** Here $\mathbf{d}$ needs to satisfy $\sum_{i=1}^{p} d_i \leq 2$. The $x$-axis shows the $l_\infty$-distance from the $L/p$ uniform dosage . We vary $p$ over $8, 9, 10$, keeping $k = 2$ and $n = 1000$ constant. $50$ different dosages are sampled at each distance, with $40$ iterations of each. Bars are $\pm.5$ std over the $2000$ squared errors at each distance.

*Figure 6.* **Misspecification.** The $x$-axis shows the $l_\infty$-distance from the half dosage. We vary $k$ from $2$ to $4$, where the true $k = 5$. We use $300$, $100$, and $200$ samples, respectively. $50$ different dosages are sampled at each distance, with $20$ iterations of each. Bars are $\pm.2$ std.

### 6.4. Extensions

In Theorem 5.1, we proved that the uniform dosage of $\frac{\mathbf{L}}{\mathbf{P}}$ is optimal in the constrained case for the simple additive models. Empirically, we see that this holds for interactive models as well, both in simulations and in numerically optimizing Eq. (4). For example, for the pairwise interaction case, Figure 5 shows the approximation error versus the deviation from the suspected optimal dosage. We see that with $L = 2$, $n = 1000$, and varying $p = 8, 9, 10$, the approximation error increases as we deviate from $\frac{L}{p}$.

### 6.5. Misspecified model

In the case where we do not know the true degree of the highest-order interaction, our model may be misspecified case. While our theoretical results do not support this case, we conduct experiments that show that the half dosage still appears to be optimal in a single-round setting. Here, we use a Boolean function of full degree (with $p = 5$), and vary $k$ between $2$ and $4$. So while the true function features interaction terms of all degrees, our assumption is that only terms of interaction up to $k$ appear in $f$. We fit the model under these assumed values of $k$, and observe that a half dosage appears to still lead to the lowest estimation errors in Figure 6.

## 7. Discussion

In this work, we propose and study probabilistic factorial design, a scalable and flexible approach to implementing factorial experiments, which generalizes both full and fractional factorial designs. Within this framework, we tackle the optimal design problem, focusing on learning combinatorial intervention models using Boolean function representations with bounded-degree interactions. We establish theoretical guarantees and near-optimal desgin strategies in both passive and active learning settings. In the passive setting, we prove that a uniform dosage of $1/2$ per treatment is near-optimal for estimating any $k$-way interaction model. In the active setting, we propose a numerically optimizable acquisition function and demonstrate its theoretical near-optimality. Additionally, we extend our approach to account for practical constraints, including limited supply, heteroskedastic multi-round noise, and emulating target combinatorial distributions. Finally, these theoretical results are validated through simulated experiments.

**Limitations and Future Work.** This work has several limitations and assumptions that may be interesting to address in future work. First, we assume a product infection mechanism in the probabilistic design. However, this assumption may not hold in certain scenarios, such as when interference or censoring effects are present. For example, in cell biology, experiments conducted on tissue samples may exhibit spatial interactions among neighboring cells. Additionally, certain treatment combinations may induce cell death, leading to a lack of observable units for those combinations. Second, our combinatorial intervention model could be extended to incorporate unit-specific covariates. The current model assumes that outcomes are determined solely by the received treatment, which suffices for homogenous units and average effects. However, incorporating covariate-based models would enable finer-grained personalized treatment-outcome predictions. Third, while we explore several extensions to the design problem, further investigations into alternative constraints, such as sparse interventions, and alternative objectives, such as optimizing specific outcome variables, could be valuable directions for future work.

## Impact Statement

This paper presents theoretical work whose goal is to advance the field of Machine Learning. There are many potential societal consequences of our work, none which we feel must be specifically highlighted here.

## Acknowledgements

We thank the anonymous reviewers for helpful comments. D.S. was supported by the Advanced Undergraduate Research Opportunities Program at MIT. J.Z. was partially supported by an Apple AI/ML PhD Fellowship. C.U. was partially supported by NCCIH/NIH (1DP2AT012345), ONR (N00014-22-1-2116 and N00014-24-1-2687), the United States Department of Energy (DE-SC0023187), the Eric and Wendy Schmidt Center at the Broad Institute, and a Simons Investigator Award.

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

# A. Proof for Section 3

## A.1. Fourier Representation

**Lemma A.1.** $f$ admits the representation $f(\mathbf{x}) = \sum_{S \subseteq [p]} \beta_S \phi_S(\mathbf{x})$, where $\beta_S = \frac{1}{2^p} \sum_{\mathbf{y} \in \{-1,1\}^p} f(\mathbf{y}) \phi_S(\mathbf{y})$.

*Proof.* Plugging in the value of $\beta_S$, we have

$$
\begin{aligned}
f(\mathbf{x}) &= \sum_{S \subseteq [p]} \left( \frac{1}{2^p} \sum_{\mathbf{y} \in \{-1,1\}^p} f(\mathbf{y}) \phi_S(\mathbf{y}) \right) \phi_S(\mathbf{x}) \\
&= \frac{1}{2^p} \sum_{S \subseteq [p]} \sum_{\mathbf{y} \in \{-1,1\}^p} f(\mathbf{y}) \prod_{i \in S} x_i y_i \\
&= \frac{1}{2^p} \sum_{\mathbf{y} \in \{-1,1\}^p} f(\mathbf{y}) \sum_{S \subseteq [p]} \prod_{i \in S} x_i y_i \\
&= \frac{1}{2^p} f(\mathbf{x}) 2^p \\
&= f(\mathbf{x}),
\end{aligned}
$$

$\square$

as $\sum_{S \subseteq [p]} \prod_{i \in S} x_i y_i = 2^p$ if $\mathbf{x} = \mathbf{y}$ and 0 otherwise.

**Lemma A.2.** *Consider a model on $\{-1,1\}^p$, where $\psi_S(\mathbf{x}) = 1$ iff $\mathbf{x}_i = 1$ for all $i \in S$, and*

$$
g(\mathbf{x}) = \sum_{S \in [p]} \alpha_S \psi_S(\mathbf{x}).
$$

*This model is a specific case of our model, where a low-interaction constraint on this model implies a low-interaction constraint on our model.*

*Proof.* We have

$$
\begin{aligned}
g(\mathbf{x}) &= \sum_{S \subseteq [p]} \alpha_S \psi_S(\mathbf{x}) \\
&= \sum_{S \subseteq [p]} \alpha_S \prod_{i \in S} \frac{(x_i + 1)}{2} \\
&= \sum_{S \subseteq [p]} \frac{1}{2^{|S|}} \alpha_S \sum_{T \subseteq S} \prod_{i \in T} x_i \\
&= \sum_{T \subseteq [p]} \left( \sum_{S \supseteq T} \frac{\alpha_S}{2^{|S|}} \right) \phi_T(\mathbf{x}).
\end{aligned}
$$

Therefore,

$$
g(\mathbf{x}) = \sum_{S \subseteq [p]} \beta_S \phi_S(\mathbf{x}),
$$

where

$$
\beta_S = \sum_{T \supseteq S} \frac{\alpha_T}{2^{|T|}}.
$$

Note that $\alpha_S = 0$ for all $|S| > k$ implies that $\beta_S = 0$ for all $|S| > k$.

$\square$

## B. Proofs for Section 4

### B.1. Properties of $\Sigma(d)$

**Lemma B.1.** *Let $\Sigma(\mathbf{d}) = \mathbb{E}\phi(\mathbf{x})^T\phi(\mathbf{x})$, where $\phi(\mathbf{x})$ is the row vector composed of $\phi_S(\mathbf{x})$ for all $S$ with $|S| \leq k$ and $\mathbf{x}$ is distributed according to the dosage $\mathbf{d}$. Then the minimum eigenvalue of $\Sigma(\mathbf{d})$ is at most $1$, with equality iff $\mathbf{d} = \frac{1}{2}\mathbf{1}_p$.*

*Proof.* First note that $\Sigma(\mathbf{d})$ is given by

$$\Sigma(\mathbf{d})_{S,S'} = \prod_{i \in S \Delta S'} (2d_i - 1)$$

Therefore, $\Sigma$ is symmetric with diagonal elements equal to $1$. In addition, $\Sigma(\mathbf{d})$ is positive semidefinite and hence has real, non-negative eigenvalues. Combined with the fact that the trace of $\Sigma(\mathbf{d})$ is $M$, the mean of the eigenvalues must be $1$. Therefore, the minimum eigenvalue is equal to $1$ if and only if all the eigenvalues are equal to $1$. A real symmetric matrix has a spectrum of only $1$'s if and only if it is the identity. Noting that $\Sigma(\mathbf{d})_{\emptyset,\{i\}} = 2d_i - 1$, $\Sigma(\mathbf{d}) = \mathbf{I}_K$ if and only if $d_i = \frac{1}{2}$, concluding the proof. $\square$

**Lemma B.2.** *With $\Sigma(\mathbf{d})$ defined as above, we have $\lambda_{\min}(\Sigma(\mathbf{d})) \leq \min_i(1 - |2d_i - 1|)$.*

*Proof.* We proceed with proof by contradiction. Let $c^* = \min_i(1 - |2d_i - 1|)$ and $i^* = \operatorname{argmin}_i(1 - |2d_i - 1|)$. If $\lambda_{\min}(\Sigma(\mathbf{d})) > c^*$, then $\Sigma(\mathbf{d}) - c^*\mathbf{I}_K$ is positive definite because $\Sigma(\mathbf{d})$ is positive semidefinite. Therefore, all leading principal minors of $\Sigma(\mathbf{d}) - c^*\mathbf{I}_K$ must have positive determinants. Consider the $2 \times 2$ submatrix defined by the rows/columns corresponding to $\emptyset$ and $\{i^*\}$). In $\Sigma - c^*\mathbf{I}_K$, this is $\begin{bmatrix} |2d_i^* - 1| & 2d_i^* - 1 \\ 2d_i^* - 1 & |2d_i^* - 1| \end{bmatrix}$, which has determinant $0$. Note that this submatrix is a principal minor in a permuted version of $\Sigma - c^*I$, which is also positive definite. Therefore, we have a contradiction as $\Sigma - c^*\mathbf{I}_K$ is not positive definite, and hence $\lambda_{\min}(\Sigma(\mathbf{d})) \leq \min_i(1 - |2d_i - 1|)$. $\square$

### B.2. Proof of Lemma 4.1

**Lemma B.3** (Truncated OLS). *Given a fixed design matrix $\mathcal{X}$, the truncated OLS estimator satisfies the following property:*

$$\mathbb{E}_Y\left[\|\hat{\boldsymbol{\beta}} - \boldsymbol{\beta}\|_2^2\right] = \begin{cases} \sum_{i=1}^K \frac{\sigma^2}{\lambda_i(\mathcal{X}^\top\mathcal{X})}, & \text{if } \sum_{i=1}^K \frac{1}{\lambda_i(\mathcal{X}^\top\mathcal{X})} \leq \frac{B^2}{\sigma^2}, \\ \|\boldsymbol{\beta}\|_2^2, & \text{otherwise.} \end{cases}$$

*In particular, there is $\min\{\sum_{i=1}^K \frac{\sigma^2}{\lambda_i(\mathcal{X}^\top\mathcal{X})}, \|\boldsymbol{\beta}\|_2^2\} \leq \mathbb{E}\left[\|\hat{\boldsymbol{\beta}} - \boldsymbol{\beta}\|_2^2\right] \leq \min\{\sum_{i=1}^K \frac{\sigma^2}{\lambda_i(\mathcal{X}^\top\mathcal{X})}, B^2\}$.*

*Proof.* We utilize the eigen-decomposition $UDU^T$ of $\mathcal{X}^T\mathcal{X}$. We have

$$\begin{aligned}
\mathbb{E}\left[\left\|\hat{\boldsymbol{\beta}}^{OLS} - \boldsymbol{\beta}\right\|^2\right] &= \mathbb{E}\left[\|(\mathcal{X}^T\mathcal{X})^{-1}\mathcal{X}^T\epsilon\|^2\right] \\
&= \mathbb{E}\left[\epsilon^T\mathcal{X}(\mathcal{X}^T\mathcal{X})^{-1}(\mathcal{X}^T\mathcal{X})^{-1}\mathcal{X}^T\epsilon\right] \\
&= \mathbb{E}\left[\operatorname{tr}(\epsilon^T\mathcal{X}(\mathcal{X}^T\mathcal{X})^{-1}(\mathcal{X}^T\mathcal{X})^{-1}\mathcal{X}^T\epsilon)\right] \\
&= \operatorname{tr}\left[\mathbb{E}\left[\epsilon\epsilon^T\right]\mathcal{X}(\mathcal{X}^T\mathcal{X})^{-1}(\mathcal{X}^T\mathcal{X})^{-1}\mathcal{X}^T\right] \\
&\leq \sigma^2\operatorname{tr}\left[\mathcal{X}(\mathcal{X}^T\mathcal{X})^{-1}(\mathcal{X}^T\mathcal{X})^{-1}\mathcal{X}^T\right] \\
&= \sigma^2\sum_{i=1}^K \frac{1}{\lambda_i(\mathcal{X}^T\mathcal{X})}
\end{aligned}$$

Therefore, if $\sum_{i=1}^K \frac{1}{\lambda_i(\mathcal{X}^T\mathcal{X})} \leq \frac{B^2}{\sigma^2}$, we use the OLS estimator which has an MSE of $\sum_{i=1}^K \frac{\sigma^2}{\lambda_i(\mathcal{X}^T\mathcal{X})}$. Otherwise, if $\sum_{i=1}^K \frac{1}{\lambda_i(\mathcal{X}^T\mathcal{X})} > \frac{B^2}{\sigma^2}$, our estimator is $\mathbf{0}_K$ which has a squared error of $\|\beta\|_2^2 \leq B^2$. This gives the desired result.

$\square$

**Lemma B.4** (OLS+Ridge estimator)**.** *Given a fixed $n \times K$ design matrix $\mathcal{X}$, the OLS+Ridge estimator is defined by*

$$\hat{\boldsymbol{\beta}} = \begin{cases} \hat{\boldsymbol{\beta}}^{OLS} & \frac{1}{\lambda_{\min}(\mathcal{X}^T\mathcal{X})} \leq \frac{B^2 n}{B^2 \lambda_{\min}(\mathcal{X}^T\mathcal{X}) + Kn\sigma^2}, \\ \hat{\boldsymbol{\beta}}^{ridge} & otherwise. \end{cases}$$

*and satisfies*

$$\mathbb{E}_Y\left[\|\hat{\boldsymbol{\beta}} - \boldsymbol{\beta}\|_2^2\right] \leq \min\left(\frac{K\sigma^2}{\lambda_{\min}(\mathcal{X}^T\mathcal{X})}, \frac{B^2 Kn\sigma^2}{B^2 \lambda_{\min}(\mathcal{X}^T\mathcal{X})^2 + Kn\sigma^2}\right). \tag{9}$$

*Proof.* The bound on OLS follows easily from the proof of Proposition B.3, where we have that

$$\sigma^2 \sum_{i=1}^{K} \frac{1}{\lambda_i(\mathcal{X}^T\mathcal{X})} \leq \frac{K\sigma^2}{\lambda_{\min}(\mathcal{X}^T\mathcal{X})}.$$

Now, we analyze the ridge estimator. Recall the definition:

$$\hat{\boldsymbol{\beta}}^{ridge} = (\mathcal{X}^T\mathcal{X} + \lambda\mathbf{I}_K)^{-1}\mathcal{X}^T y$$

where $\lambda$ is a chosen regularization parameter. The bias-variance decomposition gives us

$$\mathbb{E}\left[\left\|\hat{\boldsymbol{\beta}}^{ridge} - \boldsymbol{\beta}\right\|^2\right] = \left\|\mathbb{E}\left[\hat{\boldsymbol{\beta}}^{ridge}\right] - \boldsymbol{\beta}\right\|^2 + \mathbb{E}\left[\left\|\hat{\boldsymbol{\beta}}^{ridge} - \mathbb{E}\left[\hat{\boldsymbol{\beta}}^{ridge}\right]\right\|^2\right].$$

We analyze each term separately. For the bias term, we have

$$\mathbb{E}\left[\hat{\boldsymbol{\beta}}^{ridge} - \boldsymbol{\beta}\right] = ((\mathcal{X}^T\mathcal{X} + \lambda\mathbf{I}_K)^{-1}\mathcal{X}^T\mathcal{X} - \mathbf{I}_K)\boldsymbol{\beta}$$
$$= -\lambda(\mathcal{X}^T\mathcal{X} + \lambda\mathbf{I}_K)^{-1}\boldsymbol{\beta}$$

so that

$$\left\|\mathbb{E}\left[\hat{\boldsymbol{\beta}}^{ridge} - \boldsymbol{\beta}\right]\right\|^2 = \lambda^2 \boldsymbol{\beta}^T(\mathcal{X}^T\mathcal{X} + \lambda\mathbf{I}_K)^{-1}(\mathcal{X}^T\mathcal{X} + \lambda\mathbf{I}_K)^{-1}\boldsymbol{\beta}$$
$$\leq \lambda^2 \|\boldsymbol{\beta}\|^2 \max_{\|x\|=1}\left\|(\mathcal{X}^T\mathcal{X} + \lambda\mathbf{I}_K)^{-1}x\right\|^2$$
$$= \lambda^2 \|\boldsymbol{\beta}\|^2 \lambda_{\max}((\mathcal{X}^T\mathcal{X} + \lambda\mathbf{I}_K)^{-1}(\mathcal{X}^T\mathcal{X} + \lambda\mathbf{I}_K)^{-1})$$
$$= \frac{\lambda^2 \|\boldsymbol{\beta}\|^2}{(\lambda_{\min}(\mathcal{X}^T\mathcal{X}) + \lambda)^2}.$$

Now for the variance, we have (where once again, we use the eigen-decomposition $\mathcal{X}^T\mathcal{X} = UDU^T$)

$$\mathbb{E}\left[\left\|\hat{\boldsymbol{\beta}}^{ridge} - \mathbb{E}\left[\hat{\boldsymbol{\beta}}^{ridge}\right]\right\|^2\right] = \mathbb{E}\left[\left\|(\mathcal{X}^T\mathcal{X} + \lambda\mathbf{I}_K)^{-1}\mathcal{X}^T\epsilon\right\|^2\right]$$
$$= \mathbb{E}\left[\epsilon^T\mathcal{X}(\mathcal{X}^T\mathcal{X} + \lambda\mathbf{I}_K)^{-1}(\mathcal{X}^T\mathcal{X} + \lambda\mathbf{I}_K)^{-1}\mathcal{X}^T\epsilon\right]$$
$$= \mathbb{E}\left[\operatorname{tr}(\epsilon^T\mathcal{X}(\mathcal{X}^T\mathcal{X} + \lambda\mathbf{I}_K)^{-1}(\mathcal{X}^T\mathcal{X} + \lambda\mathbf{I}_K)^{-1}\mathcal{X}^T\epsilon)\right]$$
$$= \operatorname{tr}\left[\mathbb{E}\left[\epsilon\epsilon^T\right]\mathcal{X}(\mathcal{X}^T\mathcal{X} + \lambda\mathbf{I}_K)^{-1}(\mathcal{X}^T\mathcal{X} + \lambda\mathbf{I}_K)^{-1}\mathcal{X}^T\right]$$
$$\leq \sigma^2\operatorname{tr}\left[\mathcal{X}(\mathcal{X}^T\mathcal{X} + \lambda\mathbf{I}_K)^{-1}(\mathcal{X}^T\mathcal{X} + \lambda\mathbf{I}_K)^{-1}\mathcal{X}^T\right]$$
$$= \sigma^2\operatorname{tr}\left[UD(D + \lambda\mathbf{I}_K)^{-1}(D + \lambda\mathbf{I}_K)^{-1}U^T\right]$$
$$= \sigma^2\operatorname{tr}\left[D(D + \lambda\mathbf{I}_K)^{-1}(D + \lambda\mathbf{I}_K)^{-1}\right]$$
$$= \sigma^2 \sum_{i=1}^{K} \frac{\lambda_i(\mathcal{X}^T\mathcal{X})}{(\lambda_i(\mathcal{X}^T\mathcal{X}) + \lambda)^2}$$
$$\leq \frac{\sigma^2\operatorname{tr}(\mathcal{X}^T\mathcal{X})}{(\lambda_{\min}(\mathcal{X}^T\mathcal{X}) + \lambda)^2}.$$

With the knowledge that $\|\beta\|_2^2 \leq B^2$, we choose

$$\lambda = \frac{\sigma^2 \text{tr}(\mathcal{X}^T \mathcal{X})}{B^2 \lambda_{\min}(\mathcal{X}^T \mathcal{X})}$$

which gives us an overall bound of

$$\frac{B^2 \sigma^2 \text{tr}(\mathcal{X}^T \mathcal{X})}{B^2 \lambda_{\min}(\mathcal{X}^T \mathcal{X})^2 + \sigma^2 \text{tr}(\mathcal{X}^T \mathcal{X})}.$$

$\square$

**Lemma B.5** (Concentration of $\frac{1}{n} \mathcal{X}^T \mathcal{X}$)**.** *Let $\mathcal{X}$ be the (random) design matrix generated by dosage $\mathbf{d}$. Then*

$$\mathbb{P}\left(\left\|\frac{1}{n} \mathcal{X}^T \mathcal{X} - \Sigma(\mathbf{d})\right\| \leq t\right) \geq 1 - 2\exp\left(K \ln 9 - \frac{nt^2}{8K^2}\right)$$

*where the first norm is the spectral norm, and $\Sigma(\mathbf{d})$ is defined as in Lemma B.1.*

*Proof.* This proof loosely follows the proof of Theorem 4.5.1 in (Vershynin, 2018). Let $\mathcal{X}$ be the (random) design matrix generated by dosage $\mathbf{d}$. Recall that $\mathcal{X}^T \mathcal{X}$ is a $K \times K$ matrix. Now, let $\mathcal{N}$ be a $\frac{1}{4}-$ net on the unit sphere $S^{K-1}$ with $|\mathcal{N}| \leq 9^K$ (Vershynin, 2018). We have

$$\left\|\frac{1}{n} \mathcal{X}^T \mathcal{X} - \Sigma(\mathbf{d})\right\| \leq 2 \max_{x \in \mathcal{N}} \left|\left\langle \left(\frac{1}{n} \mathcal{X}^T \mathcal{X} - \Sigma(\mathbf{d})\right) x, x\right\rangle\right| = 2 \max_{x \in \mathcal{N}} \left|\frac{1}{n} \|\mathcal{X}x\|_2^2 - x^T \Sigma(\mathbf{d}) x\right| \tag{10}$$

where the first norm is the spectral norm. This chain of inequalities follows from the definition of an $\epsilon-$net and the triangle inequality (Vershynin, 2018). Let $\mathcal{X}_i$ denote the $i$th row of $\mathcal{X}$, and define $Z_i := \langle \mathcal{X}_i, x\rangle$. Then we have $\|\mathcal{X}x\|^2 = \sum_{i=1}^n \langle \mathcal{X}_i, x\rangle^2 = \sum_{i=1}^n Z_i^2$ where $Z_i^2 \leq K$ by Cauchy-Schwarz. It follows that $|Z_i^2 - x^T \Sigma(\mathbf{d}) x| \leq 2K$, as $\mathbb{E}[Z_i^2] = x^T \Sigma(\mathbf{d}) x$. Therefore, by Hoeffding's inequality, we have that

$$\mathbb{P}\left(\left|\frac{1}{n} \sum_{i=1}^n Z_i^2 - x^T \Sigma(\mathbf{d}) x\right| \geq t\right) \leq 2\exp\left(-\frac{nt^2}{2K^2}\right).$$

Now, applying union bound over $\mathcal{N}$ and substituting into (9), we have

$$\mathbb{P}\left(\left\|\frac{1}{n} \mathcal{X}^T \mathcal{X} - \Sigma\right\| \leq t\right) \geq 1 - 9^K \cdot 2\exp\left(-\frac{nt^2}{8K^2}\right)$$

$$= 1 - 2\exp\left(K \ln 9 - \frac{nt^2}{8K^2}\right).$$

$\square$

## B.3. Proof of Theorem 4.2

*Proof.* We have that, under the half dosage,

$$n\mathbb{E}\left[\left\|\hat{\beta} - \beta\right\|^2\right] \leq n \min\{\sum_{i=1}^K \frac{\sigma^2}{\lambda_i(\mathcal{X}^\top \mathcal{X})}, B^2\}$$

$$\leq \mathbb{P}\left(\left\|\frac{1}{n} \mathcal{X}^\top \mathcal{X} - \Sigma(\mathbf{d})\right\| \leq \delta\right) \sum_{i=1}^K \frac{\sigma^2}{\left(\frac{\lambda_{\min}(\mathcal{X}^T \mathcal{X})}{n}\right)} + \mathbb{P}\left(\left\|\frac{1}{n} \mathcal{X}^\top \mathcal{X} - \Sigma(\mathbf{d})\right\| > \delta\right) nB^2$$

$$\leq \frac{K\sigma^2}{1 - \delta} + nB^2 \exp\left(K \ln 9 - \frac{n\delta^2}{8K^2}\right) \tag{11}$$

where we use Lemma B.1, Lemma B.5, and Weyl's inequality ($|\lambda_{\min}(A) - \lambda_{\min}(B)| \leq \|A - B\|$ for real, symmetric $A, B$) in the last step.

Next, we lower bound 4 for any other dosage $\mathbf{d}$. Let $c = \min_i(1 - |2d_i - 1|)$. We have

$$
\begin{aligned}
n\mathbb{E}\left[\left\|\hat{\beta} - \beta\right\|^2\right] &\geq n \min\{\sum_{i=1}^K \frac{\sigma^2}{\lambda_i(\mathcal{X}^\top \mathcal{X})}, \|\beta\|^2\} \\
&\geq \mathbb{P}\left(\left\|\frac{1}{n}\mathcal{X}^\top \mathcal{X} - \Sigma(\mathbf{d})\right\| \leq \delta\right) \min\{\sum_{i=1}^K \frac{\sigma^2}{\lambda_i(\Sigma(\mathbf{d})) + \delta}, n\|\beta\|^2\} \\
&\geq \left(1 - 2\exp\left(K \ln 9 - \frac{n\delta^2}{8K^2}\right)\right) \min\{\sum_{i=1}^K \frac{\sigma^2}{\lambda_i(\Sigma(\mathbf{d})) + \delta}, n\|\beta\|^2\} \\
&\geq \left(1 - 2\exp\left(K \ln 9 - \frac{n\delta^2}{8K^2}\right)\right) \min\{\frac{\sigma^2}{c + \delta} + \frac{\sigma^2(K-1)}{1 + \delta}, n\|\beta\|^2\} \quad (12)
\end{aligned}
$$

where the last step is by Lemma B.2 and Cauchy-Schwarz:

$$
\left(\sum_{i=1}^K \frac{1}{\lambda_i(\Sigma(\mathbf{d})) + \delta}\right)\left(\sum_{i=1}^K \lambda_i(\Sigma(\mathbf{d})) + \delta\right) \geq K^2 \quad (13)
$$

with equality if and only if $\lambda_i(\Sigma(\mathbf{d}))$ are equal for all $i$. In particular, because $\lambda_{\min}(\Sigma(\mathbf{d})) \leq c$ by Lemma B.2, we have that

$$
\sum_{i=1}^K \frac{1}{\lambda_i(\Sigma(\mathbf{d})) + \delta} \geq \frac{1}{c + \delta} + \sum_{i=2}^K \frac{1}{\lambda_i(\Sigma(\mathbf{d})) + \delta} \geq \frac{1}{c + \delta} + \frac{K - 1}{1 + \delta}
$$

applying Cauchy-Schwarz as we did in (13).

Setting $\delta = \delta_1$ in 11 and $\delta = \delta_2$ in 12, the $\frac{1}{2}$ dosage is optimal to within a factor of

$$
\frac{\frac{K\sigma^2}{1 - \delta_1} + nB^2 \exp\left(K \ln 9 - \frac{n\delta_1^2}{8K^2}\right)}{\left(1 - 2\exp\left(K \ln 9 - \frac{n\delta_2^2}{8K^2}\right)\right) \min\{\frac{K\sigma^2}{1 + \delta_2}, n\|\beta\|^2\}}
$$

which is the result of dividing expression 11 by expression 12, and plugging in $c = 1$ in 12. We further have that for $n$ large enough, if

$$
c < \sigma^2 \left(\frac{\frac{K\sigma^2}{1 - \delta_1} + nB^2 \exp\left(K \ln 9 - \frac{n\delta_1^2}{8K^2}\right)}{1 - 2\exp\left(K \ln 9 - \frac{n\delta_2^2}{8K^2}\right)} - \frac{\sigma^2(K - 1)}{1 + \delta_2}\right)^{-1} - \delta_2
$$

then $\mathbf{d}$ results in a lower mean squared error than the $\frac{1}{2}$ dosage. This expression is the result of solving for $c$ such that expression 12 is greater than expression 11.

Choosing $\delta_1 = \delta_2 = \left(\frac{2\ln(n)}{n}\right)^{1/2}$, we have optimality to a factor of

$$
1 + O\left(\frac{\ln(n)}{n}\right)
$$

and that the optimal solution $\mathbf{d}^*$ must satisfy $c \geq 1 - O\left(\sqrt{\frac{\ln(n)}{n}}\right)$, i.e.

$$
\left\|\mathbf{d}^* - \frac{1}{2}\right\|_\infty < O\left(\sqrt{\frac{\ln(n)}{n}}\right).
$$

$\square$

This result can be extended to allow for arbitrary distributions over combinations, rather than product distributions over treatments as induced by dosages:

**Theorem B.6.** *Allowing for any distribution over combinations, the uniform distribution over combinations is optimal to a factor of at most $1 + O\left(\frac{\ln(n)}{n}\right)$. In particular, as $n \to \infty$, the uniform distribution minimizes the mean squared error of the truncated OLS estimator.*

*Proof.* The same argument used to show Lemma B.1 can be extended to arbitrary distributions. Let $\Sigma(g) = \mathbb{E}\phi(\mathbf{x})^T \phi(\mathbf{x})$ where $\phi(\mathbf{x})$ is distributed according to the distribution $g$ over combinations. Then, once again, $\Sigma(g)$ has trace $K$ for all $g$, and $\sum_{i=1}^{K} \frac{\sigma^2}{\lambda_i(\Sigma(g))}$ is minimized when $\Sigma(g)$ is the identity matrix (refer to the Cauchy-Schwarz argument above). This is achieved by $g = \mathcal{U}(\{-1, 1\}^p)$, i.e. the uniform distribution over combinations. Therefore, we may repeat the argument above to get the same result on the optimality factor of the uniform distribution in this more general case. $\square$

### B.4. Proof of Theorem 4.4

*Proof.* Let $P = \frac{1}{n} \sum_{i=1}^{t-1} \mathcal{X}_i^T \mathcal{X}_i$, where $\mathcal{X}_i$ is the design matrix from round $i$. Now, define

$$\mathbf{d}^* = \operatorname*{argmin}_{d} \sum_{i=1}^{K} \frac{1}{\lambda_i(\Sigma(\mathbf{d}) + P)}.$$

We begin by showing an upper bound on 4 when the design matrix at round $t$ is generated by $\mathbf{d}^*$. Let $\mathcal{X}$ denote the cumulative design matrix after $t$ rounds, so that $\mathcal{X}^T \mathcal{X} = \mathcal{X}_t^T \mathcal{X}_t + nP$. We have

$$
\begin{aligned}
n\mathbb{E}\left[\left\|\hat{\beta} - \beta\right\|^2\right] &\leq n \min\{\sum_{i=1}^{K} \frac{\sigma^2}{\lambda_i(\mathcal{X}^T \mathcal{X})}, B^2\} \\
&\leq \sum_{i=1}^{K} \frac{\sigma^2}{\left(\frac{\lambda_{\min}(\mathcal{X}^T \mathcal{X})}{n}\right)} + \mathbb{P}\left(\left\|\left(\frac{1}{n}\mathcal{X}_t^\top \mathcal{X}_t + P\right) - (\Sigma(\mathbf{d}^*) + P)\right\| > \delta\right) nB^2 \\
&\leq \sum_{i=1}^{K} \frac{\sigma^2}{\lambda_i(\Sigma(\mathbf{d}^*) + P) - \delta} + nB^2 \exp\left(K \ln 9 - \frac{n\delta^2}{8K^2}\right)
\end{aligned}
\tag{14}
$$

Where we use Lemma B.5 and Weyl's inequality, as in the proof of Theorem 4.2.

Next, we lower bound 4 for any other dosage $\mathbf{d}$. We have

$$
\begin{aligned}
n\mathbb{E}\left[\left\|\hat{\beta} - \beta\right\|^2\right] &\geq n \min\{\sum_{i=1}^{K} \frac{\sigma^2}{\lambda_i(\mathcal{X}^\top \mathcal{X})}, \|\beta\|^2\} \\
&\geq \mathbb{P}\left(\left\|\frac{1}{n}\mathcal{X}^\top \mathcal{X} - (\Sigma(\mathbf{d}) + P)\right\| \leq \delta\right) \min\{\sum_{i=1}^{K} \frac{\sigma^2}{\lambda_i(\Sigma(\mathbf{d}) + P) + \delta}, n\|\beta\|^2\} \\
&\geq \left(1 - 2\exp\left(K \ln 9 - \frac{n\delta^2}{8K^2}\right)\right) \min\{\sum_{i=1}^{K} \frac{\sigma^2}{\lambda_i(\Sigma(\mathbf{d}) + P) + \delta}, n\|\beta\|^2\}
\end{aligned}
\tag{15}
$$

Setting $\delta = \delta_1$ in 14 and $\delta = \delta_2$ in 15, the $\mathbf{d}^*$ is optimal to within a factor of

$$
\frac{\sum_{i=1}^{K} \frac{\sigma^2}{\lambda_i(\Sigma(\mathbf{d}^*) + P) - \delta_1} + nB^2 \exp\left(K \ln 9 - \frac{n\delta_1^2}{8K^2}\right)}{\left(1 - 2\exp\left(K \ln 9 - \frac{n\delta_2^2}{8K^2}\right)\right) \min\{\sum_{i=1}^{K} \frac{\sigma^2}{\lambda_i(\Sigma(\mathbf{d}) + P) + \delta_2}, n\|\beta\|^2\}}
$$

which is the result of dividing expression 14 by expression 15.

Choosing $\delta_1 = \delta_2 = \left( \frac{2 \ln(n)}{n} \right)^{1/2}$, we have optimality of $\mathbf{d}^*$ to a factor of at most

$$1 + O\left( \frac{\ln(n)}{n} \right).$$

$\square$

## C. Proof for Section 5

### C.1. Proof of Theorem 5.1

*Proof.* Following the proof of theorems 4.2, and noting that both terms in Eq. (9) are decreasing in $\lambda_{\min}(\mathcal{X}^T \mathcal{X})$, it suffices to show that among dosages satisfying $\sum_{i=1}^{p} d_i \leq L$, the uniform dosage with values $\frac{L}{p}$ leads to $\Sigma(\mathbf{d})$ with the highest minimum eigenvalue. When $k = 1$, $\Sigma(\mathbf{d})$ can be written as below. Let $c_i = 1 - (2d_i - 1)^2$. Define the following two matrices: $y$ is the $p$−length column vector with $y_i = 2d_i - 1$, and $C$ is the diagonal matrix with $C_{ii} = c_i$. Then

$$\Sigma(\mathbf{d}) = \begin{bmatrix} 1 & y^T \\ y & yy^T + C \end{bmatrix}$$

We compute $\det(\Sigma(\mathbf{d}) - \lambda \mathbf{I}_K)$ using the formula for the determinant of a block matrix and the matrix determinant lemma. Let $c_i = 1 - (2d_i - 1)^2$. Then for $\lambda \neq 1, c_i$ for any $i$, we have

$$\det(\Sigma(\mathbf{d}) - \lambda \mathbf{I}_K) = (1 - \lambda) \det(yy^T + C - \lambda \mathbf{I}_K - \frac{1}{1 - \lambda} yy^T)$$

$$= (1 - \lambda) \prod_{i=1}^{p} (c_i - \lambda) \left[ 1 - \frac{\lambda}{1 - \lambda} \sum_{i=1}^{p} \frac{1 - c_i}{c_i - \lambda} \right]$$

Therefore, the eigenvalues can be 1, $c_i$ for any $i$, or the solutions to $1 - \frac{\lambda}{1-\lambda} \sum_{i=1}^{p} \frac{1-c_i}{c_i - \lambda} = 0$. Define

$$g_{\mathbf{d}}(\lambda) = 1 - \frac{\lambda}{1 - \lambda} \sum_{i=1}^{p} \frac{1 - c_i}{c_i - \lambda}$$

where $c_i$'s are defined according to $\mathbf{d}$.

WLOG, assume $c_1 \leq c_2 \ldots \leq c_p$. We first note that the minimum eigenvalue must lie in $[0, c_1)$, as $g_{\mathbf{d}}(\lambda)$ must have a root in this interval. This is because $g(0) = 1$ (unless the $d_i = 0$ or 1 for some $i$, in which case $\Sigma(\mathbf{d})$ is singular) and $\lim_{\lambda \to c_1} g(\lambda) = -\infty$.

Now, let $\mathbf{d}^*$ be the uniform dosage with elements $\frac{L}{p}$, so that $c^* = 1 - \left( \frac{2L}{p} - 1 \right)^2$. The minimum eigenvalue here is given by

$$\lambda^* = \frac{1}{2} \left( c^* + 1 + p(1 - c^*) - \sqrt{(c^* + 1 + p(1 - c^*))^2 - 4c^*} \right),$$

so it suffices to show that for any dosage (satisfying the constraint) that

$$c_1 > \lambda^* \Rightarrow g_{\mathbf{d}}(\lambda^*) \leq 0,$$

implying that there is a root to $g_{\mathbf{d}}(\lambda)$ that is less than or equal to $\lambda^*$.
Note that $g_{\mathbf{d}^*}(\lambda^*) = 0$, so it suffices to show that

$$c_1 > \lambda^* \Rightarrow g_{\mathbf{d}}(\lambda^*) \leq g_{\mathbf{d}^*}(\lambda^*).$$

Now, treating $g$ as a function of $\mathbf{c} = (c_1, c_2, \ldots c_p)$ parametrized by $\lambda$, it suffices to show that $g$ is concave in $\mathbf{c}$. This would imply that a maximizer exists at a uniform dosage (since $g$ is symmetric in $\mathbf{c}$), and that dosage must be $\frac{L}{p}$ as $h(c) = \frac{1-c}{c-\lambda}$ is decreasing in $c$. We have concavity of $g$ as the Hessian is a diagonal matrix with the $i$th element being $\frac{-2\lambda^*}{(c_i - \lambda^*)^3} \leq 0$ as $c_1 > \lambda^*$. Therefore, the minimum eigenvalue of $\Sigma(\mathbf{d})$ is indeed maximized at $\mathbf{d}^*$, and we may proceed with the proof as in Theorem 4.2. $\square$

## D. Experiment Details

Code can be found at the linked repository. Below we give a few additional details of our experiments.

**Hardware and libraries.** Experiments were run on a device with a 16 core Intel Core Ultra 7 165H processor with 32 GB RAM, and an NVIDIA RTX 4000 Mobile Ada Generation 12 GB GPU. The code is implemented in Python, utilizing the `cupy` and `numba` libraries, among others. The active design optimization was done using `scipy SLSQP` solver.

