# OpenReview forum: "Probabilistic Factorial Experimental Design for Combinatorial Interventions"
_ICML.cc/2025/Conference — ICML 2025 spotlightposter_

### Official Review · Reviewer_jf24 · 2025-03-14

**Overall Recommendation:** 4

**Summary:**

This paper studies the combinatorial intervention problem. The authors propose a probabilistic factorial experimental design, where each unit independently receives a random combination of treatments according to specified dosages. They derive a closed-form solution for the near-optimal design in the passive setting and a numerically optimizable solution for the near-optimal design in the active setting. Simulation results are provided to validate their findings.

**Claims And Evidence:**

They are generally well-supported to me. The simulation results clearly align with the theories.

**Essential References Not Discussed:**

I don't see any obvious missing references.

**Experimental Designs Or Analyses:**

The experimental designs are generally sound. However, the paper could be strengthened by including a sensitivity analysis that varies $k$.

**Methods And Evaluation Criteria:**

The simulations are carefully designed to validate the theoretical results. For example, the stated near-optimality of setting the dosage to 1/2 for each treatment in the passive setting is clearly demonstrated in Figures 1 and 2. Additionally, the simulations for the active setting illustrate the effectiveness of adaptively choosing the dosages in accordance with the proposed theory.

**Other Comments Or Suggestions:**

The x-axis in Figure 2 seems to be the dosage value rather than $\\|d-\frac{1}{2}\\|_{\infty}$.

**Other Strengths And Weaknesses:**

Strengths: The paper is generally well written and smooth to follow. Then extensions are enlightening.
Weaknesses: Simulation results on real-world dataset are missing.

**Questions For Authors:**

According to Figures 1 and 2, $\mathbf{d} = (1 / 2, \ldots, 1 / 2)$ appears to be exactly optimal, rather than merely near-optimal. Do the authors conjecture that this is indeed optimal for general $k$? Similarly, in the general constrained case, is the uniform dosage of $L / p$ conjectured to be optimal?

**Relation To Broader Scientific Literature:**

The proposed probabilistic factorial design includes both full and fractional factorial designs in the literature as special cases. It serves as a flexible realization of a factorial design.

**Theoretical Claims:**

The proofs appear sound to me.

---

> ### Author Rebuttal · Authors · 2025-04-01
>
> We thank the reviewer for finding our extensions enlightening. We find the reviewer's suggestions insightful and accordingly lay out additional experiments and their results.
>
> > Simulation results on real-world dataset are missing.
>
> We thank the reviewer for sharing this concern. Our paper is concerned with how an experimenter could optimally construct a dataset through choice of dosage, and therefore, experiments with real-world data necessitate close collaboration with individuals engaged in experimentation. However, we could have simulated the performance of various dosages using a real-world dataset, if this dataset included samples with all combinations. For example, for each combination generated by a given dosage, we could draw a corresponding point from the dataset to construct a new dataset consistent with the dosage. Unfortunately, we are not aware of such complete datasets. We plan to collaborate with biological experimenters in the future to create datasets based on our results.
>
> While existing real-world datasets are not feasible for experimentation, we added a semi-synthetic simulation in the following way: we use a real-world Boolean function with $p=5$, which is a reliability function originally presented in Quality and Reliability Engineering International of full-degree [1]. We create datasets based on this function, where each sample corresponds to a combination (according to the dosage) and the corresponding value of the function, with noise added. We conduct experiments using this function, results of which can be found in tabular format below. We note that $p=5$ is relatively low, but we were unable to find a completely-defined Boolean function of higher dimension.
>
> > However, the paper could be strengthened by including a sensitivity analysis that varies k.
>
> We thank the reviewer for this suggestion. We have accordingly conducted the following experiment: we use a real-world Boolean function with $p=5$ and of full degree (described above). We replicate the first experiment in our paper, where we investigate the effect of $||\mathbf{d}-\frac{1}{2}||_\infty$. We investigate values of $k$ ranging from $2$ through $4$. We display the average loss ($\mathbb{E}_x\left[||f(x) - \hat{f}(x)||_2^2\right]$) over $200$ dosages at each distance (where we perform $20$ trials with each dosage), to be displayed in graphical format in our paper. Here we use $n=200$ samples.
>
>
> || $0$ | $.02$ | $.04$ | $.06$ | $.08$ | $.1$ | $.12$ | $.14$ | $.16$ | $.18$ |
> | -- | -- | -- |--| -- | -- | -- |--| -- | -- | -- |
> |$k=2$     |$3.49$|$3.50$|$3.53$|$3.58$|$3.64$|$3.71$|$3.81$|$3.98$|$4.12$|$4.27$|
> | $k=3$     |$.59$|$.59$|$.61$|$.62$|$.64$|$.67$|$.74$|$.78$|$.87$|$1.11$|
> | $k=4$     |$.58$|$.58$|$.72$|$.65$|$.95$|$1.17$|$1.39$|$1.73$|$1.90$|$2.30$|
>
>
> Even when the model is misspecified, we see that the half dosage appears optimal and observe the loss increase as we move further away from the half dosage.
>
> > The x-axis in Figure 2 seems to be the dosage value rather than $||d-\frac{1}{2}||_\infty$
>
> We thank the reviewer for catching this, which we will fix accordingly in the paper.
>
> > According to Figures 1 and 2, $\bf{d}=(\frac{1}{2}, \ldots , \frac{1}{2})$ appears to be exactly optimal, rather than merely near-optimal. Do the authors conjecture that this is indeed optimal for general k? Similarly, in the general constrained case, is the uniform dosage of $L/p$ conjectured to be optimal?
>
> Based on our experiments and the general heuristic that in linear regression, one would like features which are "spread out," we conjecture that the half dosage is optimal and that the uniform dosage in the constrained case is also optimal for general $k$. It is difficult to prove exact optimality, as we must compare the quantity  $\mathbb{E}_{\mathcal{X}}\left[\sum\_{i=1}^K \frac{1}{\lambda_i(\mathcal{X}^T\mathcal{X})}\right]$ across different dosages. While we were able to show that the inner quantity concentrates as the number of samples grows, it is not clear to us how to compute the mean for a fixed number of samples.
>
> ---
> Reference:
> $[1]$ Montgomery, Douglas C. Design and analysis of experiments. John Wiley & Sons, 2017.

---

### Official Review · Reviewer_CWrH · 2025-03-15

**Overall Recommendation:** 3

**Summary:**

This paper introduces probabilistic factorial experimental design for combinatorial interventions, where each treatment is assigned a dosage between 0 and 1, and units randomly receive treatments based on these probabilities.
This framework generalizes both full and fractional factorial designs by allowing random assignment of treatment combinations rather than deterministic selection.
The authors model outcomes using Boolean functions with Fourier expansions to capture bounded-order interactions.
They prove that uniform half-dosage allocation ($d_i = 0.5$) is near-optimal in single-round experiments, with optimality up to a factor of $1 + O(\frac{\operatorname{ln}(n)}{n})$. For multi-round experiments (i.e., active learning setting), they develop an acquisition strategy that adapts dosages based on previous observations. The work also addresses practical constraints like limited treatment supply. Experiments results on simulated datasets demonstrate that the proposed strategies outperform random dosage selection.

**Claims And Evidence:**

The paper's claims are generally well-supported by theoretical analysis and empirical evidence.

**Essential References Not Discussed:**

All the essential related works are discussed.

**Experimental Designs Or Analyses:**

I checked the experimental designs in Section 6, both the passive setting and active setting simulations. The authors appropriately test the theoretical claims by comparing estimation errors across different dosage strategies. I have some minor concerns:

1. The simulations use synthetic data generated from the same model class assumed in the theory, but this might not reflect the robustness of the proposed framework to model misspecification.

2. In the active setting, the authors could include more existing active learning methods as baselines beyond random and half-dosage strategies for a more comprehensive evaluation.

**Methods And Evaluation Criteria:**

The methods and evaluation criteria in this paper are appropriate for the problem of optimal experimental design for combinatorial interventions. My only concern is that all the evaluations are conducted on synthetic data. It would be great if the authors could conduct some experiments on semi-synthetic or real-world datasets.

**Other Comments Or Suggestions:**

Please see Other Strengths And Weaknesses.

**Other Strengths And Weaknesses:**

Strengths: The theoretical analysis in this paper is rigorous and well-organized, making it easy to understand the theoretical results and their practical implications.

Weaknesses:

1. Please see my comments in the previous parts.

2. The computational complexity of the active learning approach is not thoroughly discussed.

3. The empirical results suggest that the optimal acquisition strategy only outperforms the half strategy slightly in the active setting. This raises questions about whether the half strategy might be preferable in practice since it requires no computation or learning procedure. A more thorough discussion of this trade-off between computational complexity and performance gain would strengthen the paper.

**Questions For Authors:**

please see Other Strengths And Weaknesses.

**Relation To Broader Scientific Literature:**

This paper extends classical factorial design literature by introducing a probabilistic framework that addresses scalability issues in traditional full and fractional factorial designs. The active learning component relates to Bayesian experimental design and sequential experimental design, though with acquisition functions specific to the probabilistic factorial framework. The work also complements recent advances in causal inference for combinatorial interventions and provides theoretical foundations for experimental practices used in biological perturbation experiments.

**Theoretical Claims:**

I checked the proof of Theorem 4.2, which establishes the near-optimality of half-dosage allocation. The proof appears sound, using concentration inequalities and eigenvalue properties to bound the estimation error.

---

> ### Author Rebuttal · Authors · 2025-04-01
>
> We thank the reviewer for appreciating our theoretical analysis, as well as for their many valuable suggestions. Below, we address the reviewer's concerns and lay out modifications we will make according to the reviewer's suggestions.
>
> > My only concern is that all the evaluations are conducted on synthetic data. It would be great if the authors could conduct some experiments on semi-synthetic or real-world datasets.
>
> Due to the character limit, we refer the reviewer to our response to Reviewer jf24's first point under "Simulation results on real-world dataset are missing".
>
> > The simulations use synthetic data generated from the same model class assumed in the theory, but this might not reflect the robustness of the proposed framework to model misspecification.
>
> We thank the reviewer for pointing this out. While Boolean functions are universal approximators, our low-degree assumption can cause misspecification, as recognized by the reviewer. In many applications the low-degree assumptions holds, especially in biology, but degree misspecification may still exist. To address the reviewer's concern, we have conducted an additional experiment where the model is misspecified. Please see the response to Reviewer jf24, under the comment about "sensitivity analysis." Here, we use a real-world full-degree Boolean function ($k=p$), and fit assuming lesser values of $k$.
>
> > In the active setting, the authors could include more existing active learning methods as baselines beyond random and half-dosage strategies for a more comprehensive evaluation.
>
> We appreciate the reviewer's suggestion. For the comparison with passive baselines, we have added an additional baseline based on partial factorial design. Results are shown in the table below. For the comparison with active strategies, since multiple combinatorial interventions are drawn in each round (administered by a selection of dosage), we are not aware of existing methods that can be easily adapted to this setting. However, we would be happy to include additional baselines if the reviewer has specific suggestions.
>
> Here we compare a Resolution $V$ $2^{5-1}$ fractional design versus our optimal strategy and half dosages. Each round has $16$ samples, with $p=5$ and $k=1$.
>
> |  | Round 1 | Round 2 |Round 3| Round 4| Round 5|
> | -------- | -------- | -------- | -------- | -------- | -------- |
> | Optimal dosage     | $.54$| $\bf{.16}$|$\bf{.10}$|$\bf{.08}$|$\bf{.07}$|
> | Half dosage    |$.52$|$.23$|$.14$|$.09$|$.08$||
> |Fractional factorial design|$\bf{.28}$|$.21$|$.15$|$.12$|$.09$|
>
> Bolded entries show the lowest loss among each round, where we see that our optimal dosage strategy outperforms the other strategies after the first round.
>
> > The computational complexity of the active learning approach is not thoroughly discussed.
>
> The number of iterations for the optimizer to converge is roughly $O(p^3)$, and the complexity of each iteration is $O(nK^2 + K^3)$ (where the first term comes from the matrix multiplication of $\mathcal{X}^T\mathcal{X}$ and the second term comes from computing the eigenvalues of $\Sigma(\mathbf{d})$). Recall the definition of $K$ to be the number of interactions under consideration, i.e. $K = \sum_{i=0}^k {p\choose i} = O(p^k)$ for small $k$. Therefore, the overall complexity is $O(np^{3k+3}+p^{6k+3})$ for small $k$. In practice, we may recommend using a proxy, which only involves the inverse of the minimum eigenvalue: $\bf{d}_{T} = \text{argmin}\_{\bf{d}\in[0,1]^p}\frac{1}{\lambda\_{\min}\left(\Sigma(\bf{d})+\frac{1}{n}\sum\_{t=1}^{T-1}{\mathcal{X}_t^\top\mathcal{X}_t}\right)}$. We found that numerically optimizing this was significantly faster and that the solver was consistently accurate. While the complexity computed above should be the same for this approach, in practice it takes many less iterations to converge.
>
> > The empirical results suggest ... A more thorough discussion of this trade-off between computational complexity and performance gain would strengthen the paper.
>
> We thank the reviewer for this suggestion and will include a discussion of this trade-off in our paper. In the case where there are not many samples (compared to features) per round, we find that the optimal acquisition strategy more clearly outperforms the half strategy. This is because when we have a smaller number of samples, we will need to "correct" as the distribution of combinations will be more lopsided and further away from the uniform distribution. Therefore, in scenarios where each round has few samples, we think it is worth computing the optimal acquisition dosage. When we have a large $n$ relative to $p$, the half strategy and optimal strategy perform very similarly. While the computational complexity of finding the optimal strategy can quickly scale, in practice it only takes a matter of seconds to compute.
>
> ---
> Reference:
> $[1]$ Montgomery, Douglas C. Design and analysis of experiments. John Wiley & Sons, 2017.

---

### Official Review · Reviewer_D2ja · 2025-03-21

**Overall Recommendation:** 3

**Summary:**

This paper is concerned with the problem of experimental design in the high dimensional factorial setting where users may be administered combinations of treatments, and the aim is to administer a subset of treatments such that all combinations are recovered. The authors frame this problem in terms of the Fourier transform of boolean functions and assuming that the treatment status can be relaxed to probabilities of treatment. After this transformation the authors use tools from optimal experimental design for the selection mechanism. Extensions are provided to subsets and heteroskedastic settings. Empirical results show strong performance.

**Claims And Evidence:**

All theoretical claims made are well supported by theory provided in the paper.

**Essential References Not Discussed:**

The authors should have a broader literature review of the partial factorial design literature.

**Experimental Designs Or Analyses:**

Yes. The experiments are sound, though I would have like to seen a more complete comparison to partial factorial experiments.

**Methods And Evaluation Criteria:**

Yes, the method is quite sensible (and interesting), evaluation criterion is appropriate.

**Other Comments Or Suggestions:**

N/A

**Other Strengths And Weaknesses:**

Overall, I think this paper is a creative approach to the problem of design of factorial experiments.
My main complaint, as I mention above, is that the experimental evaluation here is severely limited.

**Questions For Authors:**

I am curious how this approach (specifically the active learning setting) interacts with adjustment using user covariates. Does this change the design considerations?

**Relation To Broader Scientific Literature:**

This paper addresses an interesting and highly relevant problem of factorial experimental design. While the problem itself dates back to Fisher, the authors provide a nice contribution to the literature.

**Theoretical Claims:**

Yes, I reviewed all proofs and they are sound to my reading.

---

> ### Author Rebuttal · Authors · 2025-04-01
>
> We thank the reviewer for appreciating our method and the thoughtful suggestions. We would like to address the concerns and questions of the reviewer as below.
>
> > The authors should have a broader literature review of the partial factorial design literature.
>
> We thank the reviewer for this suggestion. We will add the following paragraph to Section 2 to expand our discussion of the partial factorial design literature. In addition, we are happy to include any specific references the reviewer believes would further strengthen our coverage of related work.
>
> "A $2^{-m}$ fractional design is one where $2^{p-m}$ samples are used, each with a different combination [1]. These combinations are carefully selected to minimize aliasing. Aliasing occurs when, for the combinations selected, the interactions are linearly dependent [2][3]. In a full factorial design, there is linear independence so there is no confounding when the model is fit. In a fractional design, some aliasing will always occur in a full-degree model; however, methods proposed in literature select combinations such that the aliasing of important effects (i.e. degree-1 terms) does not occur [2]. With a low-degree assumption, aliasing can be avoided entirely. Fractional designs can be classified by their *resolution* (denoted by $R$), which determines which interactions can be potentially confounded. For example, a Resolution V fractional design eliminates any confounding between lower than degree-3 interactions, appropriate for degree-2 functions [4]. Of particular interest in literature are *minimum aberration designs*, which minimize the number of degree-$l$ terms aliased with degree-$R-l$ terms [5][6]."
>
> > I would have like to seen a more complete comparison to partial factorial experiments.
>
> We thank the reviewer for this suggestion. We have conducted an additional experiment, where we compare the half dosage versus a partial factorial design in the passive setting. Here, we generate a degree-$1$ Boolean function with $p=8$. We use a $2^{8-2}$ Resolution $V$ design with $64$ samples for each approach. Results are shown below, averaged over $300$ trials and with $\pm 1$ std.
>
> | Fractional design |Half dosage|
> | -------- | -------- |
> |$.14\pm .062$| $.16\pm.078$|
>
> With fewer samples, the careful selection of combinations will make a difference, so the fractional design can outperform the half dosage. But in many cases, especially in biological applications, careful selection of combinations is not possible which is why the much more flexible dosage design is preferable, as it enables the administration of an exponential number of combinations by choosing a linear number of dosages.
>
> However, in the active setting, the optimal dosage can outperform a fractional design. Please see the experiment in response to Reviewer CWrH, under "... the authors could include more existing active learning methods as baselines".
>
> > I am curious how this approach (specifically the active learning setting) interacts with adjustment using user covariates.
>
> We thank the reviewer for this question. We could assume the following setup in the passive setting: there are $m$ users with known covariates $\mathbf{c}\_i\in \mathbb{R}^l$, each of which receives the $n$ combinations determined by the dosage (so that we have a total of $mn$ samples). Assuming the covariates have a linear relationship with the outcome, i.e. $y\_i = \beta\_u\mathbf{c}\_i+f(\mathbf{x})$, then the optimal dosage in the passive setting is $\mathbf{d}\_u^* = \text{argmin}\_{\mathbf{d}\in [0,1]^p} \sum\_{i = 1}^{l+K}\frac{1}{\lambda\_i(A)}$ where $A=\begin{bmatrix}
>     \sum\_{i=1}^m\mathbf{c}\_i\mathbf{c}\_i^T&\sum\_{i=1}^m \mathbf{c}\_i\mathbf{y}(\mathbf{d})^T\\\\
>     \sum\_{i=1}^m \mathbf{y}(\mathbf{d})\mathbf{c}\_i^T&\Sigma(\mathbf{d})
> \end{bmatrix}$, with $\mathbf{y}(\mathbf{d})\_S = \prod\_{i\in S} (2d\_i-1)\in \mathbb{R}^{K}$ and $\Sigma(\mathbf{d})$ is as defined in the paper. We conjecture that $\mathbf{d}_u^*$ is still the half dosage. To extend to the active setting, the same objective is used as in the paper except $\Sigma(\mathbf{d})$ is replaced with $A$. We are happy to consider alternative models of user covariates if the reviewer has any specific suggestions.
>
> ---
> References:
>
> $[1]$ Box, George EP, William H. Hunter, and Stuart Hunter. Statistics for experimenters. Vol. 664. New York: John Wiley and sons, 1978.
>
> $[2]$ Gunst, Richard F., and Robert L. Mason. "Fractional factorial design." Wiley Interdisciplinary Reviews: Computational Statistics 1.2 (2009): 234-244.
>
> $[3]$ Mukerjee, Rahul, and CF Jeff Wu. A modern theory of factorial design. Springer Science & Business Media, 2007.
>
> $[4]$ Montgomery, Douglas C. Design and analysis of experiments. John Wiley & Sons, 2017.
>
> $[5]$ Fries, Arthur, and William G. Hunter. "Minimum aberration $2^{k–p}$ designs." Technometrics 22.4 (1980): 601-608.
>
> $[6]$ Cheng, Ching-Shui. Theory of factorial design. Boca Raton, FL, USA: Chapman and Hall/CRC, 2016.

---

### Official Review · Reviewer_bJhy · 2025-03-23

**Overall Recommendation:** 3

**Summary:**

The paper introduces a probabilistic factorial experimental design to address the optimal experimental design problem for combinatorial interventions.
The contribution of the paper:
1. The paper introduces a probabilistic factorial experimental design for a given choice of dosage vector.
2. The paper provides a closed-form solution for the near-optimal design for passive and active settings.
3. The authors explore extending the design framework to incorporate constraints and noisy scenarios.

**Claims And Evidence:**

Yes, the paper provides the theoretical proofs and empirical evidence to support the claims.

**Essential References Not Discussed:**

No

**Experimental Designs Or Analyses:**

Yes

**Methods And Evaluation Criteria:**

The authors validated the proposed approach using a simulated dataset for both passive and active settings.

**Other Comments Or Suggestions:**

No

**Other Strengths And Weaknesses:**

**Strengths:**
1. The paper addresses a significant gap in scalability challenges in factorial design with combinatorial interventions.
2. The theoretical framework is robust, with clear assumptions and derivations.

**Weaknesses:**
1. The use of Boolean functions and Fourier transforms is not new, as similar approaches have been explored in prior work, such as *Agarwal, A., Agarwal, A., and Vijaykumar, S.Synthetic Combinations: A Causal Inference Framework for Combinatorial Interventions*.

**Questions For Authors:**

1. The proposed design strategies appear to depend on the choice of dosage (may require prior knowledge from experimenters), which is a subset of the full factorial design. As a result, the outcomes of the proposed approach may be suboptimal. Could the authors elaborate more on this and discuss how to address it?

**Relation To Broader Scientific Literature:**

Building on previous work, this paper utilizes Boolean functions and Fourier transforms to establish the theoretical foundation of its approach.

**Theoretical Claims:**

Yes

---

> ### Author Rebuttal · Authors · 2025-04-01
>
> We thank the reviewer for appreciating our theoretical framework and scalability challenges it addressed. Below, we address the concerns and questions brought up by the reviewer.
>
> > The use of Boolean functions and Fourier transforms is not new, as similar approaches have been explored in prior work, such as Agarwal, A., Agarwal, A., and Vijaykumar, S.Synthetic Combinations: A Causal Inference Framework for Combinatorial Interventions.
>
> We thank the reviewer for this comment. As noted in section 2 (last paragraph), we used Boolean functions to model combinatorial interventions, as it can easily model the scenario where the outcome is mainly driven by low-order effects in the absence of higher-order interactions -- a common assumption in fractional factorial design. In addition, it captures generalized surface models (see section 3.1). However, the main contribution of the paper lies not in the use of Boolean functions, but in the proposal of the novel probabilistic experimental framework and the accompanying theoretical analysis of the dosage choice (described in detail in section 1), which, to our knowledge, has not been explored in prior work.
>
> > The proposed design strategies appear to depend on the choice of dosage (may require prior knowledge from experimenters), which is a subset of the full factorial design. As a result, the outcomes of the proposed approach may be suboptimal. Could the authors elaborate more on this and discuss how to address it?
>
> A full factorial design can be formulated within our framework. In particular, there would be $2^p$ rounds, each with $1$ sample. In order to fix the sample, the dosage would be chosen to be deterministic, i.e. $\mathbf{d}\in \\{0,1\\}^p$. In addition, when the number of samples is large enough for a full factorial design to be implemented, the half dosage is closely related to this design as the half dosage induces a uniform distribution over combinations. Therefore in such cases, the two approaches perform similarly. Per our theoretical results, we suggest the experimenter uses the half dosage, which requires no prior knowledge.

---

### Decision · Program_Chairs · 2025-05-01

**Decision:**

Accept (spotlight poster)

**Comment:**

Short summary: This paper proposes a clever probabilistic factorial experimental design to address the optimal experimental design problem for combinatorial interventions.
All reviews unanimously vote to accept this paper (either as “weak accept” or “accept”).
The reviews acknowledge that the paper addresses a key technical gap and provides a creative solution. The reviews also find the theoretical contributions solid and robust. The paper is also well-written making it easy to understand.
One key limitation identified in the reviews is the severely lacking experimental section. The future improvements of the paper and follow up work could center around more exhaustive experiments and real-world datasets and deeper discussion of computational complexity.